# Genome-scale identification of transcription factors that mediate an inflammatory network during breast cellular transformation

Zhe Ji [1,2,4], Lizhi He[1], Asaf Rotem[1,2,5], Andreas Janzer[1,6], Christine S. Cheng[2,7], Aviv Regev[2,3] & Kevin Struhl [1]

Transient activation of Src oncoprotein in non-transformed, breast epithelial cells can initiate an epigenetic switch to the stably transformed state via a positive feedback loop that involves the inflammatory transcription factors STAT3 and NF-κB. Here, we develop an experimental and computational pipeline that includes 1) a Bayesian network model (AccessTF) that accurately predicts protein-bound DNA sequence motifs based on chromatin accessibility, and 2) a scoring system (TFScore) that rank-orders transcription factors as candidates for being important for a biological process. Genetic experiments validate TFScore and suggest that more than 40 transcription factors contribute to the oncogenic state in this model. Interestingly, individual depletion of several of these factors results in similar transcriptional profiles, indicating that a complex and interconnected transcriptional network promotes a stable oncogenic state. The combined experimental and computational pipeline represents a general approach to comprehensively identify transcriptional regulators important for a biological process.

---

[1] Department of Biological Chemistry and Molecular Pharmacology, Harvard Medical School, Boston, MA 02115, USA. [2] Broad Institute of MIT and Harvard, Cambridge, MA 02142, USA. [3] Department of Biology, Howard Hughes Medical Institute and David H. Koch Institute for Integrative Cancer Research, Massachusetts Institute of Technology, Cambridge, MA 20140, USA. [4] Present address: Department of Pharmacology and Biomedical Engineering, Northwestern University, Evanston 60611 IL, USA. [5] Present address: Department of Medical Oncology and Center for Cancer Precision Medicine, Dana-Farber Cancer Institute, Boston 02215 MA, USA. [6] Present address: Bayer Pharma, Berlin 13353, Germany. [7] Present address: Department Biology, Boston University, Boston 02215 MA, USA. These authors contributed equally: Zhe Ji, Lizhi He. Correspondence and requests for materials should be addressed to K.S. (email: kevin@hms.harvard.edu)

Transcriptional regulatory proteins that bind specific DNA sequences are the major determinants for regulating gene expression programs that determine cell state and behavior[1–3]. It is therefore important to comprehensively identify the transcription factors and transcriptional regulatory circuits involved in dynamic biological processes and maintaining stable cell states. Individual experimental approaches provide specific types of information, but integration of the various datasets is necessary for comprehensive understanding. There are only a few examples in which transcription factors important for a biological process and transcriptional regulatory connections have been identified on a comprehensive basis[4–8]. None of these have been performed in the context of cellular transformation or cancer.

Transcriptional activator or repressor proteins recruit co-activator or co-repressor complexes to their target sites via protein-protein interactions, thereby altering the level of transcription by the general RNA polymerase II machinery[9]. Some co-activator and co-repressor complexes are enzymes that locally modify chromatin structure either by altering nucleosome positions, removing nucleosomes to generate accessible DNA, or modifying histones at specific residues through acetylation and methylation. Chromatin-modifying activities are important for transcriptional regulation, but they are not the major determinants of gene expression patterns due to their limited specificity for genomic DNA sequences and their widespread presence in different cell types. Nevertheless, locally altered chromatin structures represent genomic regions of transcription factor activity in vivo under the physiological conditions tested.

Accessible chromatin regions can be mapped on a genomic scale by DNase I hypersensitivity[10] or transposon-based ATAC-seq[11]. Accessible regions are typically several hundred bp in length, and they are generated by nucleosome-remodeling complexes that are recruited by the combined action of multiple DNA-binding (and associated) proteins that bind to motifs within the accessible region. This combinatorial recruitment is critical for biological specificity, because individual sequence motifs are short and hence occur very frequently throughout large mammalian genomes simply by chance[12,13]. Many DNase I hypersensitive regions are promoters or enhancers; these are distinguished by virtue of their proximity to the transcriptional initiation site and by histone modifications (e.g., tri-methylated H3-K4)[14].

The above considerations make it possible to use genome-scale chromatin accessibility maps to identify transcription factors that regulate gene expression programs during biological progresses. One approach is to search for sequence motifs enriched in differentially accessible chromatin regions[15,16]. However, dynamic chromatin accessibility and differential gene expression are not always correlated, and many functionally important transcription factors play a constitutive (i.e., non-regulated) role and will not be identified by this approach. More directly, genome-scale DNase I footprinting[17,18] and transposon-based ATAC-seq[11] can identify genomic regions protected by bound proteins. However, these footprinting maps require ~10 times more sequencing reads than hypersensitivity maps, and hence are considerably more expensive.

Transcription factors important in various biological contexts have been identified by integrating chromatin accessibility or footprinting analyses with gene expression profiles[19–21]. However, these previous integrative analyses did not comprehensively evaluate transcription factors for their role in the biological process of interest. Here, we develop an experimental and computational pipeline to comprehensively identify transcription factors and transcriptional regulatory circuits involved in a biological process. We apply this approach to an inducible model of cellular transformation in which transient activation of v-Src

oncoprotein converts a non-transformed breast epithelial cell line (MCF-10A) into a stably transformed state within 24 h[22,23]. This epigenetic switch between stable non-transformed and transformed states is mediated by an inflammatory positive feedback loop involving the transcription factors NF-κB and STAT3[23,24]. A few transcriptional regulatory circuits involved in this transformation model have been identified, and these are important in some other cancer cell types and human cancers[24–27]. However, a comprehensive analysis of transcriptional circuitry involved in this or any other model of cellular transformation has not been described.

Using this approach, we show that >40 transcription factors are important for transformation in this model system. Furthermore, although these factors have different DNA-binding specificities, they can affect the expression of a common set of genes. This suggests that cellular transformation is mediated by a highly interconnected transcriptional regulatory circuit that depends on the combined inputs of many transcription factors.

## Results

**Transcriptional regulatory modules during transformation**. To improve our initial transcriptional profiling analysis[22], we reanalyzed mRNA expression profiles during the process of transformation (0, 1, 2, 4, 8, 16, and 24 h time points in the presence of tamoxifen, which induces v-Src; Fig. 1a). Approximately 700 genes are differentially expressed with >1.5-fold change consistently in two biological replicates in at least one time-point (False Discovery Rate <0.007). These genes form four coherent clusters: continuously up-regulated; early up-regulated at 2 h; intermediate up-regulated at 12 h; continuously down-regulated (Fig. 1b). Principal component analysis indicates that the transcriptional program gradually evolves during the transformation process (Fig. 1c). As expected, differentially expressed genes are enriched in pathways strongly associated with cancer progression such as the inflammatory response, cell migration, angiogenesis, regulation of apoptosis, and cell proliferation (Fig. 1d and Supplementary Fig. 1).

**Genome-scale mapping of transcriptional regulatory regions**. Genome-scale mapping of DNase hypersensitive sites (DNase-seq)[10] of cells at 0, 6, and 24 h after tamoxifen treatment reveals ~212,423 accessible regions (an example region is shown in Fig. 2a). To further classify types of such regulatory regions[14,28], we performed ChIP-seq for 6 histone modifications at 0, 2, 12, 24, and 36 h after tamoxifen treatment (Fig. 2a). These results indicate that 12% of the open chromatin regions are located in active promoters (H3-K27ac and H3-K4me3), 25% are in active enhancers (H3-K27ac but no H3-K4me3), 19% are in primed enhancers (H3-K4me1 but no H3-K27ac or H3-K4me3), 16% are in heterochromatin (H3-K9me3) or polycomb-repressed regions (H3-K27me3), and the remaining 28% uncharacterized based on our histone modification analysis (Fig. 2b).

Chromatin accessibility and H3-K27ac levels are dynamically regulated during the transformation process, while the levels of various types of histone methylation are largely unchanged (Fig. 2c). Open chromatin regions in enhancers and heterochromatin are more likely to be dynamically regulated than open regions in promoters, and 5 times as many genomic regions show increased accessibility upon transformation as opposed to decreased accessibility (Fig. 2d). Among open acetylated regions, those showing increased accessibility during transformation tend to be more acetylated (Fig. 2e). This suggests that many chromatin changes are due to increased function of transcriptional activator proteins bound at enhancers that recruit nucleosome remodeler and histone acetylase complexes.

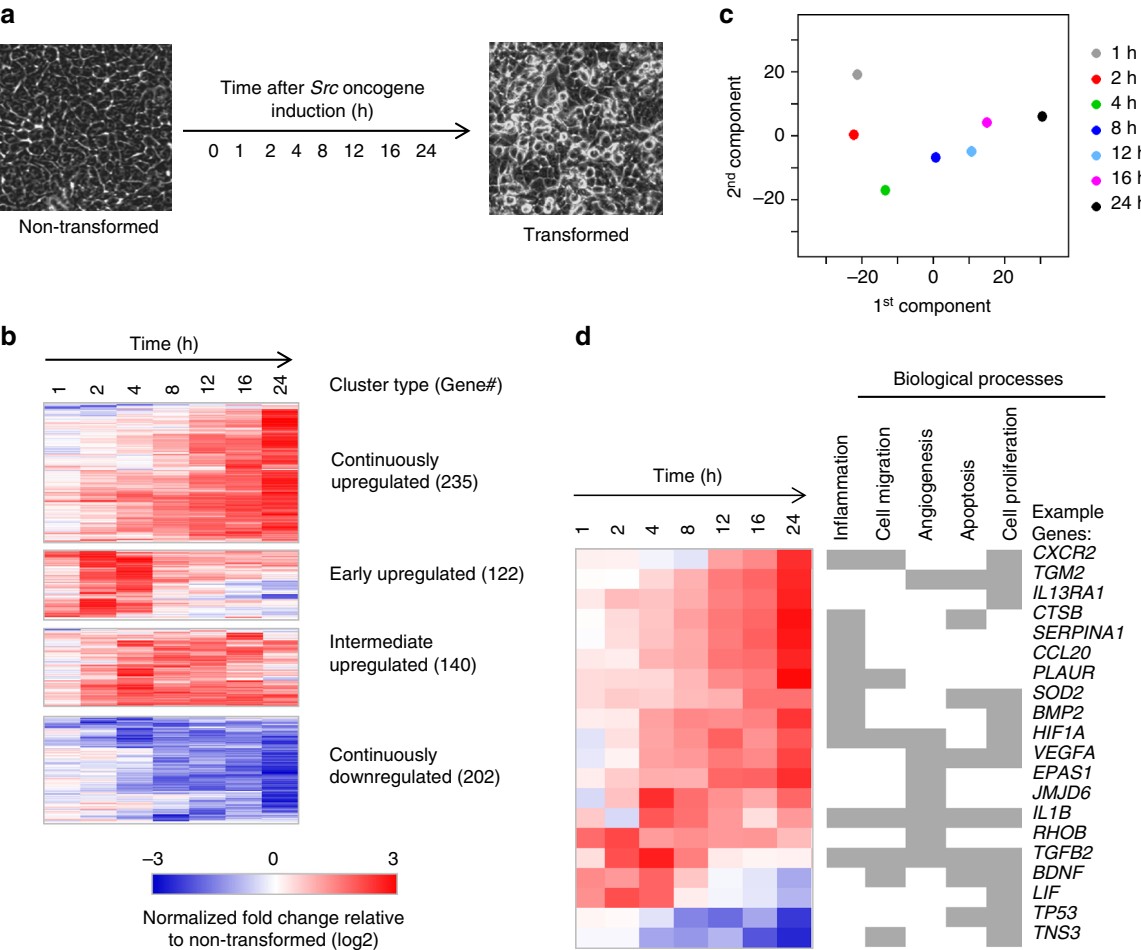

**Fig. 1** Differential gene expression during transformation. **a** Time-series of cells treated with tamoxifen for 0 h, 1 h, 2 h, 4 h, 8 h, 12 h, 16 h, and 24 h. **b** The K-mean clustering of 699 differentially expressed genes into four coherent clusters. **c** Principal component analyses of differential RNA expression profile. **d** Example genes in pathways enriched with differentially expressed genes. One gene may regulate several oncogenic processes as indicated in gray color

**Super-enhancer genes are preferentially activated**. Super-enhancers, previously termed dominant or local control regions[29], are large clusters of individual enhancers that typically drive expression of genes defining cell identity[30–32]. Using the ROSE software[30–32] and ChIP-seq data for H3-K27ac, we identified 1050 super-enhancers in at least one time point at 0, 2, and 24 h after tamoxifen treatment (Fig. 2f and Supplementary Data 1), most of which (85%) pre-exist in non-transformed cells. 367 super-enhancers (35%) show increased acetylation levels >1.5 fold after 24 h of cell transformation, whereas only 18 showed >1.5 fold decreased acetylation (Supplementary Fig. 2a). Expression of the genes located in these activated super-enhancer regions tend to be up-regulated upon transformation (Supplementary Fig. 2b). Gene ontology analyses of genes located in the super-enhancer regions are enriched in the oncogenic pathways, such as cell migration, cell proliferation, intracellular signaling cascade, angiogenesis and gene transcription (Supplementary Fig. 3).

**Bayesian network model to predict TF binding sites in vivo**. As DNase I hypersensitive regions are generated ultimately by transcription factors bound (directly or indirectly) to DNA sequences, motif analysis of these accessible regions is a straightforward approach to identify the relevant transcription factors. However, this approach involves arbitrary cut-off choices

for the quality of sequence motifs, it does not distinguish between motifs at the center or edges of accessible regions, and it does not account for different levels of accessibility. Here, we describe a Bayesian Network model approach (AccessTF) that starts from all known sequence motifs in the human genome to predict protein-binding sites in vivo from DNase I hypersensitivity data. AccessTF integrates quantitative DNase I hypersensitive measurements with the following motif information: motif quality; the distance to the closest transcription start site; conservation among vertebrates (Fig. 3a, b). We define each motif to be in a bound or unbound state, with a motif more likely to be bound if located in a DNase hypersensitive region, has higher quality, higher conservation level, and is more proximal to a transcription start site (Fig. 3b). The Bayes algorithm is converged and calculates the probability that a motif is bound.

We tested the performance of the algorithm on binding sites for AP-1 and STAT3, factors for which we have ChIP-seq data in the same cell line[33]. The Area Under ROC Curve (ROC AUC) is >0.95 for both factors, the Area Under Precision-Recall Curve (PR AUC) is 0.76 for STAT3 and 0.86 for AP-1 (Fig. 3c and Supplementary Fig. 4c), and the predicted motif binding probability increases in accord with the factor binding level (Fig. 3d and Supplementary Fig. 4b). As expected, DNase I hypersensitivity around the motif is the major parameter for distinguishing bound vs. unbound motifs, although motif information adds some power to the prediction (Fig. 3e and

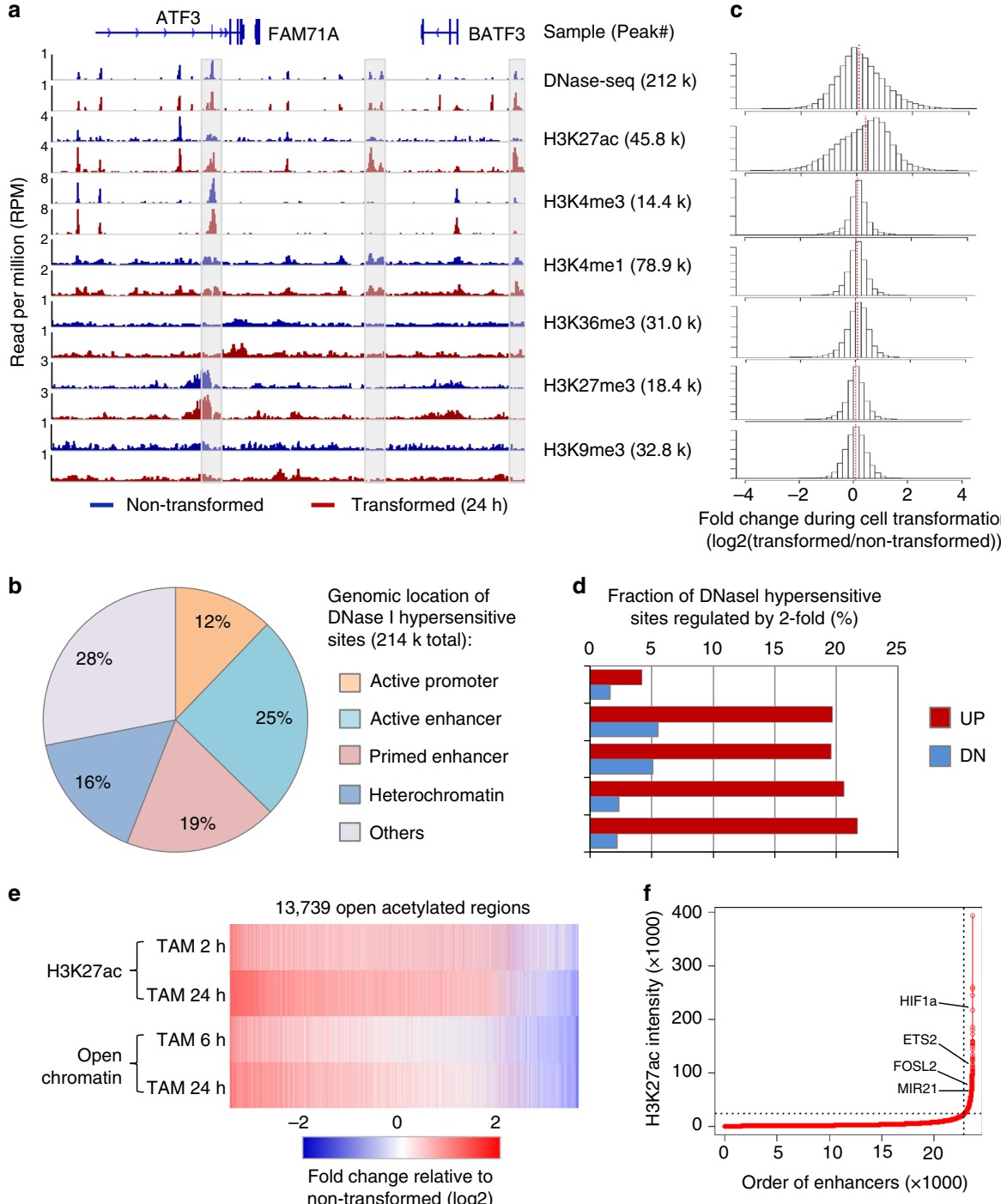

**Fig. 2** Differential chromatin accessibility and histone modification during transformation. **a** Data for the indicated chromatin features before (blue) and after (red) transformation. Total peak numbers are shown. **b** Distribution of genomic locations of DNase I hypersensitive sites (DNase-seq peaks) classified by histone modifications as active promoters (H3K4me3), active enhancers (H3K27ac), primed enhancers (H3K4me1), and heterochromatin (H3K27me3/H3K9me3). Regions without any modification were grouped as "Others". **c** Fold change of chromatin accessibility and histone modification levels, calculated as the log2 fold change of read density in cells after 24 h of tamoxifen treatment and non-transformed cells. **d** Fraction of DNase I hypersensitive sites differentially accessible (>2-fold change) during transformation. **e** Heat map showing differential chromatin accessibility and H3K27ac in 13,739 open acetylated regions during transformation. Pearson Correlation Coefficient of fold change values of accessibility vs. acetylation = 0.36 at 24 h after tamoxifen treatment; $P < 10^{-100}$. **f** Super-enhancers are identified by the sorted rank order based on H3K27ac levels at 24 h upon tamoxifen treatment. The analyses were based on the ROSE software[30–32]

Supplementary Fig. 4c). For the factors tested, the quality of the sequence motif makes a minimal contribution. As further validation of the algorithm, we performed a similar analysis in K562 cells using a DNase-seq dataset (for predictions) and ChIP-seq datasets for many transcription factors (for testing) obtained by the ENCODE consortium[34] (Supplementary Fig. 5). For many factors, the AUC curves were >0.9 and the PR AUC were >0.75, indicative of high performance. Some factors had lower AUC

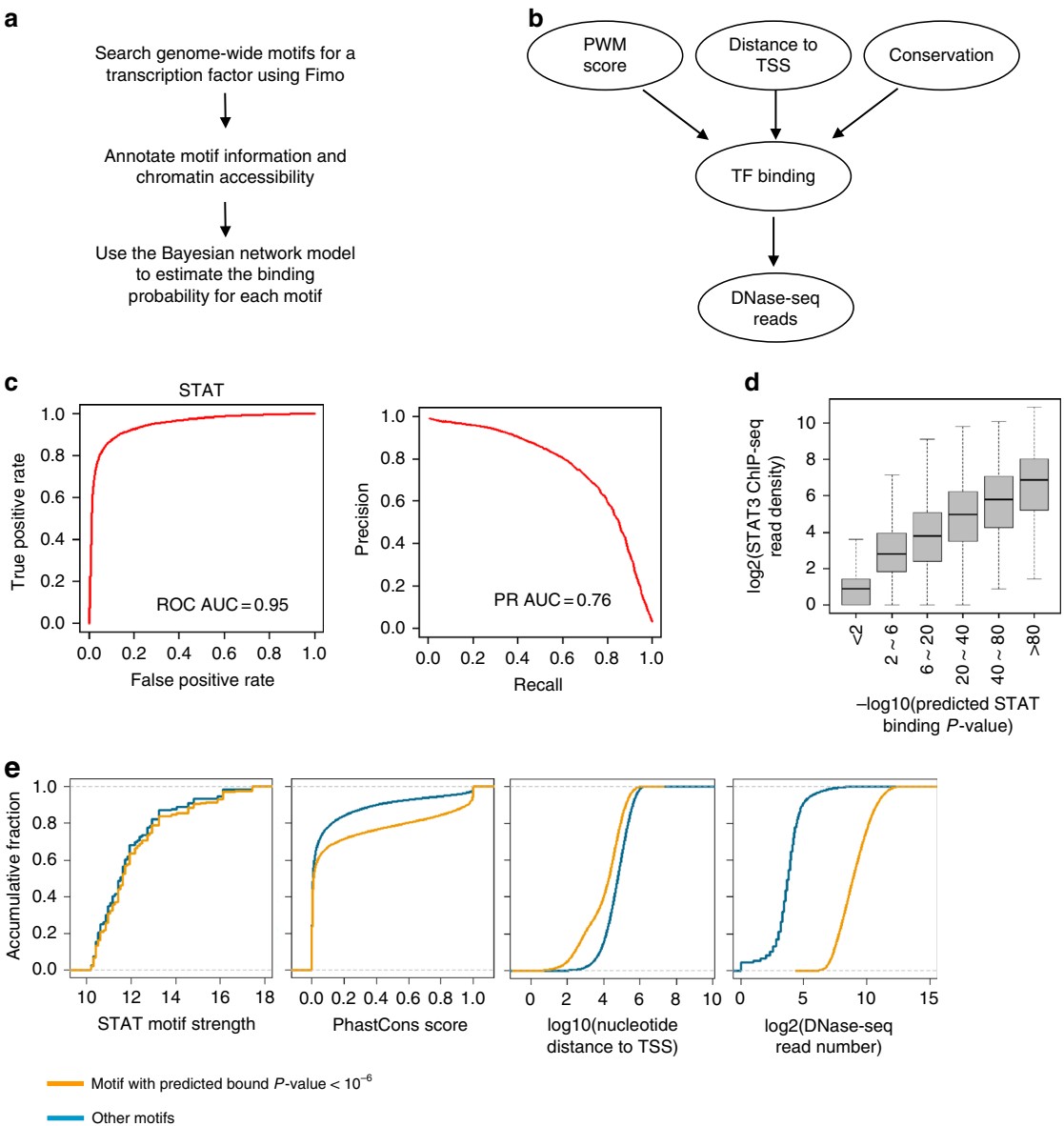

**Fig. 3** AccessTF, a Bayesian network model to identify TF binding sites. **a** Steps of computational analyses for identifying transcription factor binding sites using DNase-seq data. **b** AccessTF integrates motif strength, distance to the closest transcription start site, Phastcon conservation score and surrounding DNase-seq reads. See Methods for a detailed description. **c** Area under ROC curves (ROC AUC) and Area under Precisions-Recall Curves (PR AUC) for measuring the performance of AccessTF predicting the STAT binding status. **d** STAT motifs grouped by predicted binding probabilities and plotted against STAT3 binding levels in 400 nt region around the motifs (estimated by ChIP-seq data) after transformation. For the box plot, the bounds of the box represent the first and third quartiles and the center line represents the median. **e** Comparing features of STAT motifs with predicted bound $P$-value $< 10^{-6}$ vs. others

values (0.75–0.85), probably because these factors can bind sites that are not in open chromatin regions. Thus, the algorithm performs well to identify experimentally determined binding sites.

We then applied AccessTF to identify putative in vivo binding sites for all factors with PWM annotated by MotifDB[35]. Given the accuracy of the predictions, we expect that the vast majority of predicted sites are bound by their cognate factors in vivo under the conditions tested. However, this motif analysis does not distinguish among individual members of multi-protein families that recognize a common sequence motif (e.g., AP-1).

**Predicting TFs that regulate chromatin and expression.** To identify transcription factors important for the cellular

transformation, we examined the relative contribution of factor binding motifs, identified above, to differential chromatin accessibility and to their enrichment in promoters/enhancers of differentially expressed gene clusters. Open chromatin regions containing AP-1, NRF/MAF, STAT, and CEBP motifs are more likely to have increased accessibility during cell transformation, as compared to other open chromatin regions (Wilcoxon Rank Sum Test $p$-value $< 10^{-40}$) (Supplementary Fig. 6a). Those motifs are also enriched in promoters/enhancers of differentially expressed gene clusters (Supplementary Fig. 6b). Interestingly, these motifs are associated both with genes that are continuously up-regulated and continuously down-regulated during transformation. Such locus-specific effects on transcriptional factor function are

typically due to functional interactions with other factors that differentially bind the relevant genomic regions.

On the other hand, chromatin regions with CTCF and NFI motifs tend to show less accessibility (Wilcoxon Rank Sum Test $p$-value $< 10^{-40}$) (Supplementary Fig. 6c). Open chromatin regions with CTCF or NFI motifs and without AP-1, NRF/MAF, STAT, or CEBP motifs are even more likely to have decreased accessibility (Supplementary Fig. 6c). It is unclear whether such decreased accessibility reflects decreased activator function and hence decreased recruitment of the nucleosome remodelers or increased recruitment of transcriptional co-repressors (e.g., histone deacetylases) that inhibit the association and/or function of the remodelers.

**TFScore to predict TFs important for transformation**. We developed a score schema (TFScore) to rank transcription factors in terms of their likelihood of being important for transformation. TFScore is based on 4 criteria (Fig. 4a): 1) higher motif enrichment in promoters/enhancers of the differentially expressed gene clusters (Fig. 1b); 2) higher motif occurrences in chromatin regions showing increased accessibility at 6 and 24 h after

tamoxifen treatment (Fig. 2e); 3) up-regulation of the factor during cell transformation; 4) higher relative expression level of an individual factor of a given transcription factor family that recognize a common sequence motif. The latter two criteria were used to distinguish the contributions of the various factors with similar DNA-binding specificities, based on the idea that factors expressed at higher levels and/or up-regulated are more likely to be important for transformation. The resulting rank-ordered list of transcription factors (Fig. 4b and Supplementary Data 2) reveals known regulators STAT3 and NF-κb near the top of the list and an unexpectedly large cohort of transcriptional regulators as potentially being important for the oncogenic transformation. Although only ~10% of human protein-coding genes are present in super-enhancer regions, 24 out of the top 50 TFScore-predicted transcriptional factors are located in super-enhancer regions (Fig. 4b and Supplementary Data 2), which is highly significant (Binomial Test $P$-value $< 10^{-8}$) and suggestive of their functional importance.

**Many TFScore-predicted TFs are important for transformation**. To validate the functional importance of the predicted

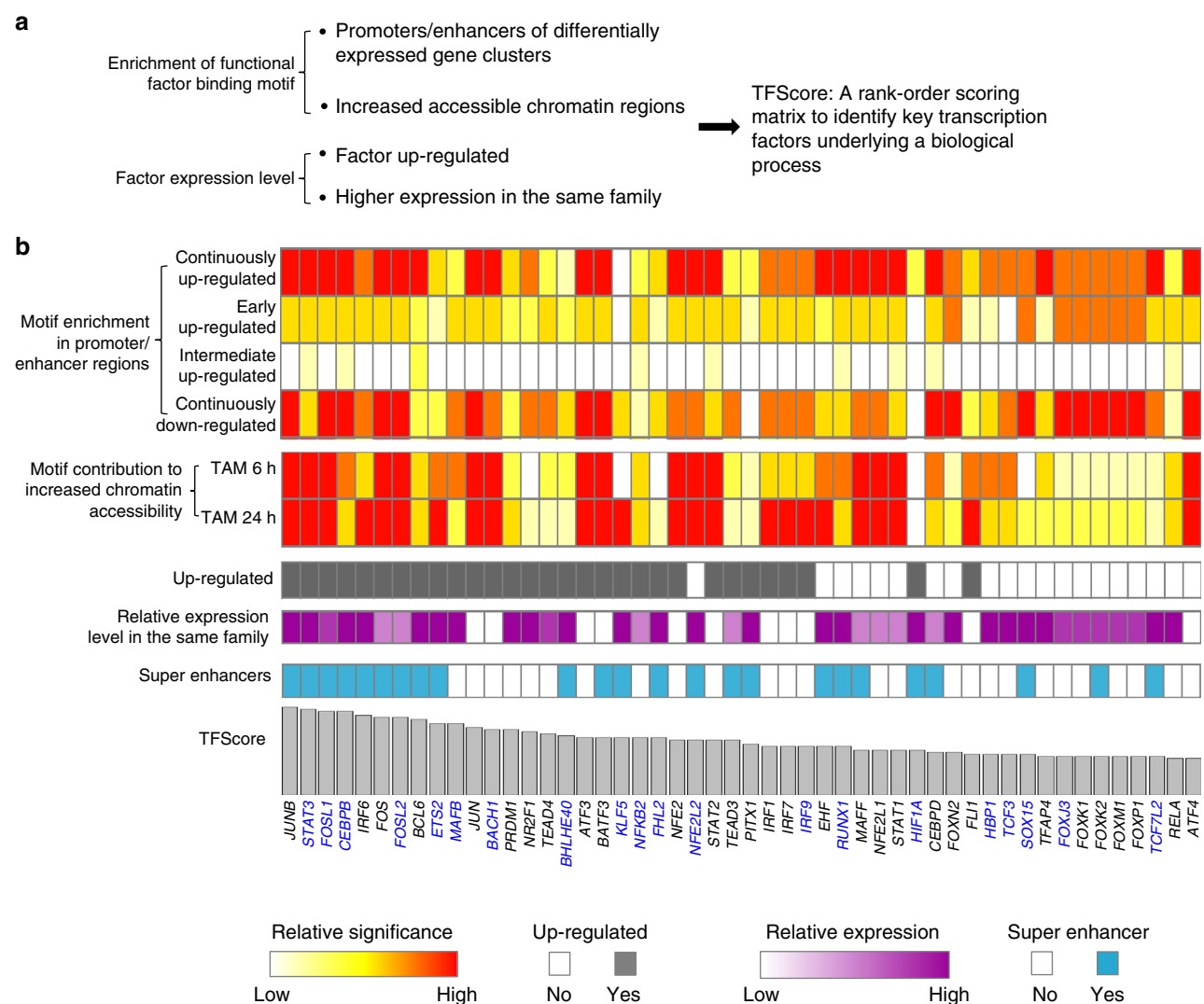

**Fig. 4** Identifying transcription factors important for transformation. **a** Criteria for TFScore. See Methods for a detailed description. **b** Rank order of transcription factors potentially playing important regulatory roles during cell transformation, based on TFScores (top 50 are shown). For each transcription factor, the relative contribution of 4 criteria to TFScore, and annotated super-enhancer genes are shown. The 20 genes used for validation by siRNA knockdowns (Fig. 5) are shown in blue

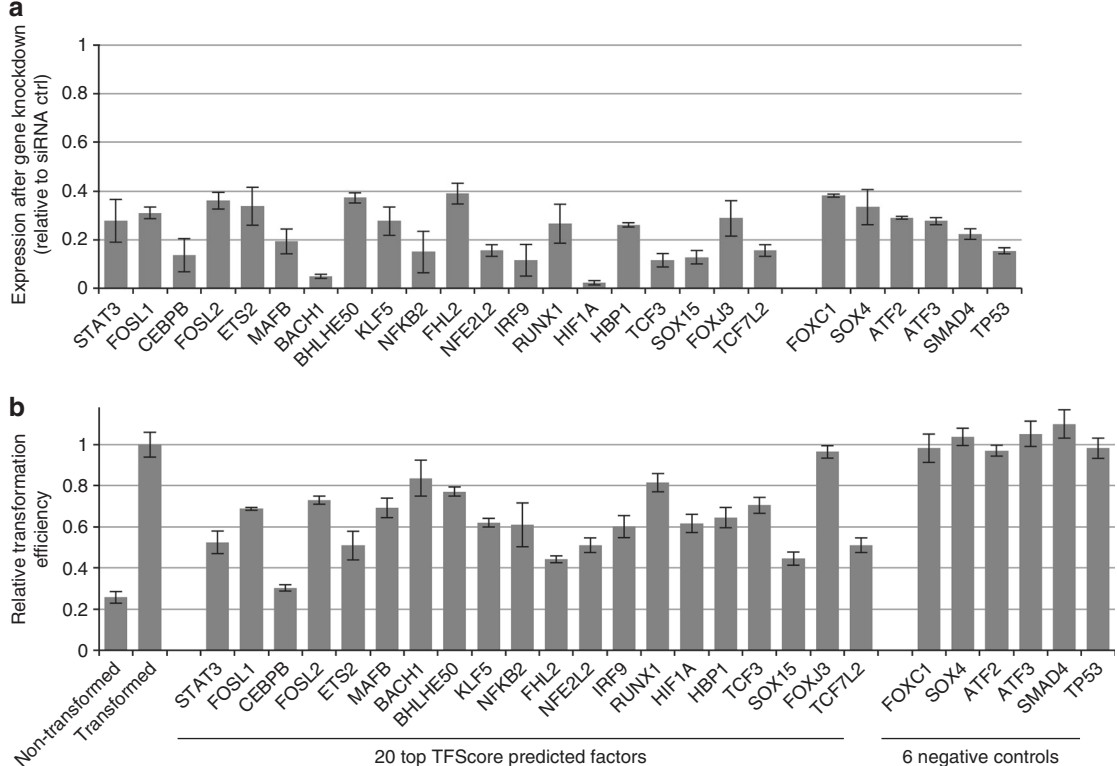

**Fig. 5** TFScore-predicted transcription factors are important for transformation. **a** siRNA knockdown efficiencies of the indicated transcription factors as compared to siRNA controls. **b** Relative transformation efficiency of the indicated transcription factors as determined by their ability to grown under low attachment conditions

transcription factors, we genetically inhibited expression of their genes via siRNAs and tested the resulting cells for their ability to grow under conditions of low attachment[36], a property of transformed cells. We randomly selected 20 of the top 50 candidate transcription factors predicted by TFScore, and as determined by mRNA levels, we obtained knockdown efficiencies of >60% (Fig. 5a). Of the 20 knockdowns tested, 17 resulted in >20% inhibition of cellular transformation and 19 resulted in >15% inhibition (Fig. 5b), indicating that the corresponding transcription factors are important for transformation. These factors include the expected STAT3 and NFKB2, but they also include FOSL1, FOSL2, CEBPB, ETS2, MAFB, BHLHE50, KLF5, FHL2, NFE2L2, IRF9, HIF1A, HBP1, TCF3, SOX15, and TCF7L2. As negative controls, knockdowns of 6 factors (FOXC1, SOX4, ATF2, ATF3, SMAD4, and TP53) with middle or low TFScores did not significantly affect cell transformation efficiencies. Thus, TFScore predicts transcription factors that are important for cellular transformation with very good accuracy with P-value < 0.0005 (Fisher's Exact Test comparing top candidate factors vs. negative controls). Some of these (TCF7L2, HIF1A, NFKB2, SOX15, FHL2, and BHLHE40) do not show high enrichment of binding sites in accessible chromatin regions that are dynamically regulated. These results indicate a surprisingly large number of factors are important for the process of cellular transformation in our model.

**TFs important for transformation co-regulate common genes**. To examine the effects of individual factors on gene expression, we performed transcriptional profiling (RNA-seq) by using siR-NAs to individually knockdown expression of 6 transcription factors (CEBPB, NFE2L2, FOSL1, FOSL2, SOX15, and TCF7L2). When normalized to a control knockdown experiment, 2576

genes show over 2-fold differential expression upon at least one factor knockdown. Remarkably, the transcriptome-scale gene expression patterns in these 6 knockdowns are quite similar, even though the factors have different DNA-binding specificities (Fig. 6a). Knockdowns of FOSL1 and FOSL2 show the most similar gene transcriptional response as compared to other knockdowns, indicating redundancy of transcription factors in the same family (Fig. 6a). For the 6 factors tested, 1428 genes (14% of the total expressed genes) are commonly down-regulated, whereas 87 (0.8% of the total expressed genes) are commonly up-regulated. The similarities in the gene expression profiles for these 6 knockdown experiments are far above random expectation (Fig. 6b). In accord with the relevance of these transcription factors to transformation, the 1428 genes commonly down-regulated upon knockdown are more likely to be up-regulated during transformation, whereas the 87 genes commonly up-regulated upon knockdown are more likely to be down-regulated during transformation (Fig. 6c).

Only 102 (7.1%) of the 1428 genes that are commonly down-regulated upon these factor knockdowns are induced >1.5 fold during cell transformation (Fig. 6a). Those genes encode important regulators of inflammatory response, cellular signaling pathways, and apoptosis (Fig. 6d). For each of these 102 inducible genes, the AccessTF-predicted binding sites in the corresponding promoter/enhancer regions provide strong evidence for which of the 6 factors directly interact with the DNA and affect transcription of the gene (on average, 4.6 AccessTF-predicted binding sites per gene). These enhancers and promoters typically lack one or more motifs, and hence direct binding sites, for transcription factors that nevertheless influence transcription of the gene. Such "non-directly-bound" transcription factors could associate with the promoter/

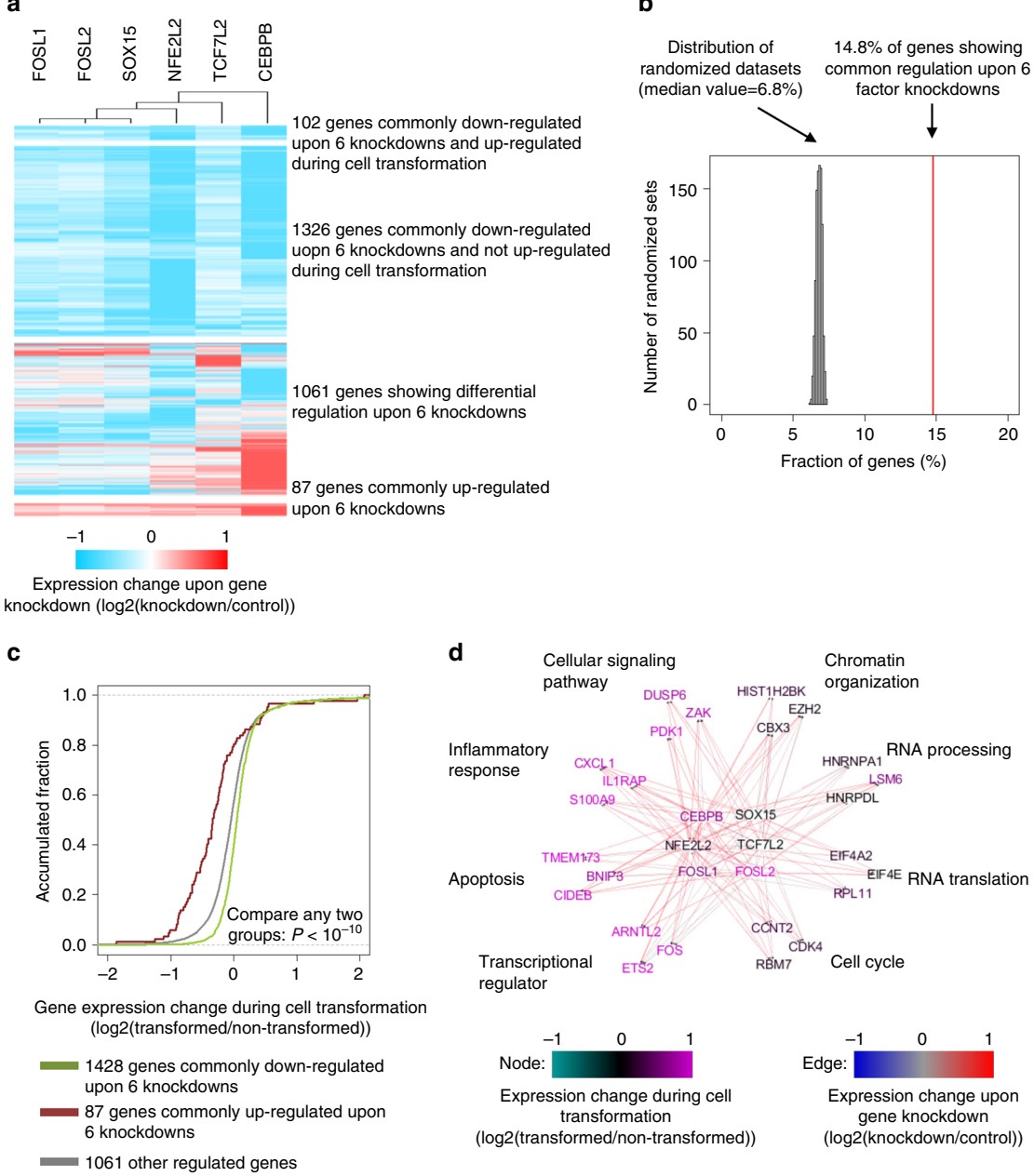

**Fig. 6** Gene expression network mediated by transcriptional regulators. **a** Genes showing significant differential gene expression upon factor knockdowns, with 2-fold differential expression in at least one knockdown. **b** Fraction of expressed genes showing consistent up- or down-regulation upon 6 factor knockdowns compared to those obtained by randomizing each column of the datasets. **c** Dynamic regulation of gene expression of gene groups in **a** during transformation. The cumulative distribution function fold changes values (log2(transformed/non-transformed)) is plotted. **d** The gene transcription network mediated by the transcription factors. The edges represent direct factor binding sites in promoters/enhancers of targeted genes. For each pathway, 3 randomly picked genes are shown

enhancer via protein-protein interactions with other factors directly bound to a different motif. Alternatively, they may indirectly affect transcription of a given gene via effects on other genes that contribute to the cell state.

We performed similar analyses for the remaining 1326 genes that are commonly down-regulated upon factor knockdowns, but are not differentially expressed during cell transformation. Gene ontology analyses show that these genes encode proteins enriched in functions such as RNA processing, cell cycle, RNA translation, and chromatin modification (Fig. 6d and Supplementary Fig. 7). Compared to inducible genes during transformation, these non-

regulated genes are less likely to be direct targets of six factors with 3.2 AccessTF-predicted binding sites per gene (Wilcoxon Rank Sum Test $P$-value $< 10^{-12}$). Interestingly, some transcription factors functionally important for cell transformation (HIF1A, ETS2, and FOS) are also common targets.

**Functional TFs and target genes are co-expressed in patients.** Using RNA-seq for the Human Cancer Cell Atlas (TCGA) database, we examined the expression of functional transcription factors and target genes learned from our cellular transformation model in breast cancer patients[37]. Expression of the top 50

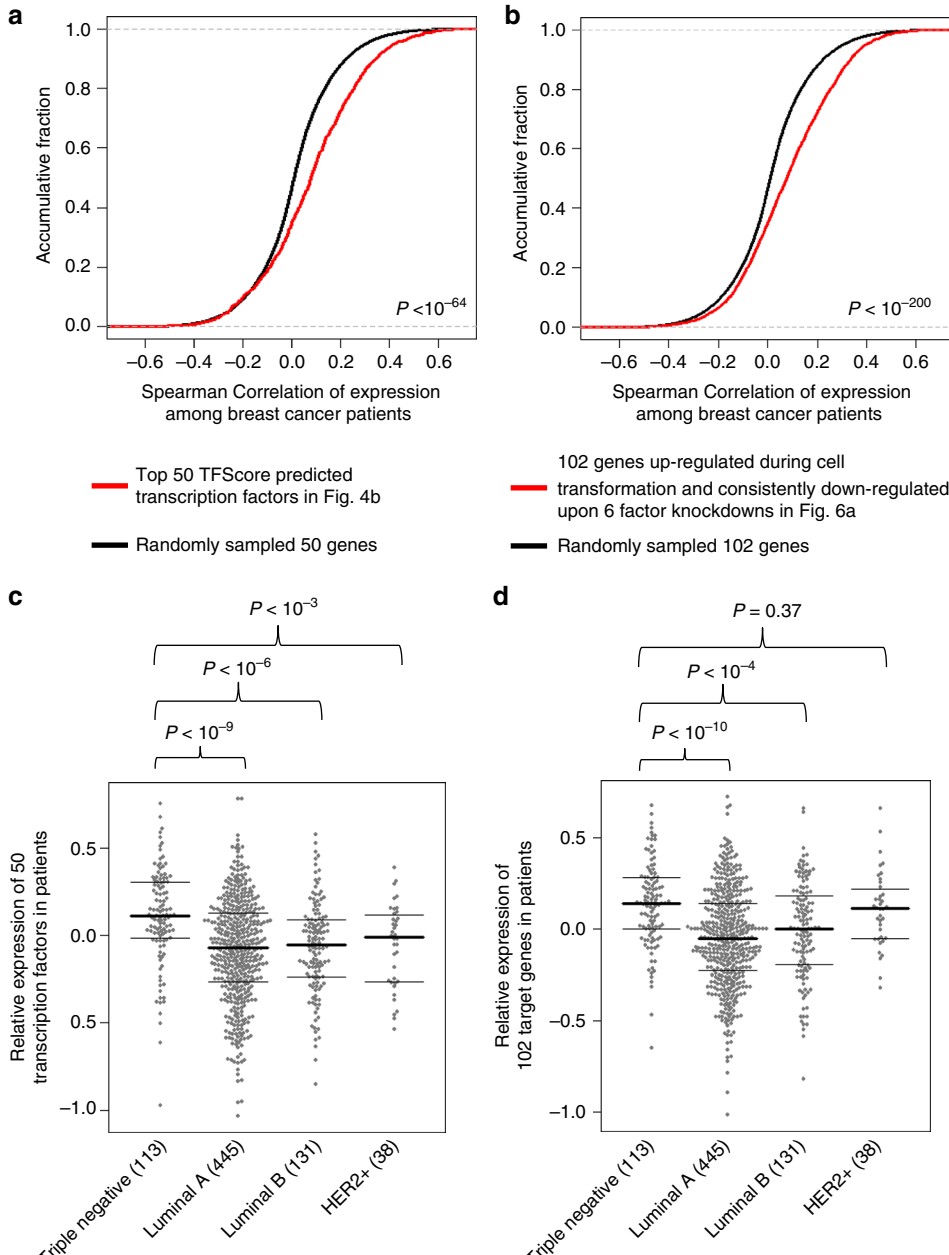

**Fig. 7** Expression of TFs and targets in 1182 breast cancer patient samples. **a** The Spearman's correlation of expression levels of top 50 TFScore predicted transcription factors in Fig. 4b. We randomly picked 50 expressed transcription factors and calculated their correlation values as the background distribution. The Wilcoxon Rank Sum test $P$-value is shown. **b** The Spearman's Correlation of expression levels of 102 genes up-regulated during cell transformation and consistently down-regulated upon 6 factor knockdowns in Fig. 5a. We randomly picked 102 expressed genes and calculated their correlation values as the background distribution. The Wilcoxon Rank Sum test $P$-value is shown. **c** Relative expression of top 50 TFScore predicted transcription factor in indicated breast cancer subtypes. The numbers of patient samples were shown in parentheses, and the Wilcoxon Rank Sum test $P$-values comparing different groups are shown. **d** Relative expression of 102 target genes in indicated breast cancer subtypes. The numbers of patient samples were shown in parentheses, and the Wilcoxon Rank Sum test $P$-values comparing different groups are shown

TFScore-predicted functional transcription factors in Fig. 4b are positively correlated among 1182 patient samples compared to randomly sampled genes (Fig. 7a). Similarly, the 102 genes that are up-regulated during cell transformation and are consistently down-regulated upon 6 factor knockdowns, show significant positive expression correlation in patient samples as compared to randomly sampled genes (Fig. 7b). In addition, the transcription factors (Fig. 7c) and target genes (Fig. 7d) identified here tend to have higher expression levels in triple negative breast cancers, as

compared to ER-positive breast cancers. Consistent with previous analyses of individual regulatory circuits in our transformation model[24–27], these data indicate that the functional transcriptional network identified in our cellular transformation model is relevant in human cancer.

## Discussion

We describe a combined experimental and computational approach to comprehensively identify transcription factors that

are important for mediating dynamic changes in gene expression between two physiological states. Experimentally, this approach combines genome-scale mapping of accessible chromatin regions via DNase I hypersensitivity, histone modifications, and transcriptional profiling. Although DNase I hypersensitive regions represent a functional assay for transcription factors bound to these regions, they do not directly identify motifs or other sequences bound by proteins in vivo. Instead, bound motifs, and hence the cognate proteins, are inferred. In contrast, genome-scale, DNase I footprinting directly identifies sequences bound by proteins in vivo[38–41]. However, while individual sequence reads contribute directly to the identification of DNase hypersensitive regions, numerous sequence reads are necessary to identify underrepresented regions of DNase I cleavage that define DNase footprints. As such, footprinting methods require much higher (~10 times more) sequencing depth and hence are considerably more expensive, especially for experiments involving multiple samples.

Computationally, we first developed a Bayesian network model, AccessTF, to predict protein-binding sites in vivo in which information about all known DNA sequence motifs is combined with quantitative measurements of DNase I hypersensitivity centered on the motifs. This approach is advantageous over standard motif analyses of accessible regions. It does not involve arbitrary cut-offs for quality of sequence motifs, it accounts for where the motif is located within the accessible region, it accounts for the level of chromatin accessibility, and it yields binding probabilities for each motif. Most importantly, validation using in vivo binding for multiple transcription factors (ChIP-seq data) yields high ROC and PR AUC values for predictions of AccessTF. Of course, any motif- or footprint-based approach cannot identify transcription factors that directly associate with target sites but instead are recruited to such sites via interactions with factors directly bound to the motif.

Secondly, we develop a novel scoring system, TFScore, to identify key transcriptional regulators by integrating the AccessTF-predicted binding sites with four layers of functional information. This integrative approach provides more powerful predictions and identifies more functional regulators, and it doesn't require a factor to meet all four criteria. TFScore yields a rank-ordered list of transcription factors that are predicted to be important for the process of interest. Experimental validation using siRNA-knockdowns indicates that most of the top 50 factors on the list are important for transformation, in contrast to all 6 factors tested with low TFScores. This high validation rate suggests that the computational pipeline should be generally applicable to identify key transcriptional regulators in other biological processes.

In principle, our integrated computational analysis is comprehensive and loosely analogous to a genetic screen, because it gives each known transcription factor a score that predicts its relative importance in transformation. Transcription factors important for cellular transformation have also been identified on a genomic scale by screening shRNA or CRISPR libraries[42,43]. This genetic approach does not identify physiological target sites or transcriptional regulatory circuits, but the identified genes are not restricted to DNA-binding transcription factors. As such, these approaches are complementary.

Our results indicate that numerous transcription factors play a functional role in transformation in a single experimental model. Extrapolation of the result that 85% of the top 50 factors (17 out of 20 tested) affect transformation suggests the involvement of at least 40 transcription factors in over 20 protein families in this oncogenic model. Moreover, it is likely that additional factors further down the list will also be important, although we have not

experimentally determined false discovery rates throughout the list. Some factors identified (e.g., NF-κB, STAT3, FOS) are known to be involved for transformation in our model, others (e.g., CEBPB, HIF1a, ETS2, FHL2, TCF7L2, and NFE2L2) have been described as oncogenes in other settings, and some proteins (BHLHE40 and MAFB) have not been well linked to cancer. Transformation in other cellular models also involves many transcription factors, although not necessarily the same set identified here[42,43]. Similarly, we hypothesize that many transcription factors will play a functional role in individual human cancers, even if only a small number of them are oncogenic drivers. The involvement of numerous transcription factors in a dynamic gene expression program has been observed in dendritic cells responding to pathogens[4,5], differentiation of Th17 cells[6,7], and hematopoiesis[8].

An important observation arising from the siRNA knockdown experiments is that the 6 factors tested affect the expression of a common group of genes. It seems likely that many of the 11 other factors validated to affect transformation will behave in a similar manner. And more broadly, as the 17 factors shown to be important for transformation were selected from and distributed among the top 50 factors, it seems likely that many of them will affect the common group of genes. This would seem to be surprising because the factors recognize different motifs, and different genes within the common group have different combinations and organizations of motifs. However, similar results have been observed in other biological processes[44–46].

Previously, we described the transformation process in our model as an epigenetic switch from a stable non-transformed state to a stable transformed state mediated by an inflammatory feedback loop[23,24]. This epigenetic switch between stable cell states is analogous to what occurs in cellular differentiation and formation of distinct and stable cell types from a common progenitor[1–3]. A similar epigenetic switch involving an inflammatory feedback loop occurs in a liver cell model of transformation[47]. STAT3 is a critical player in both epigenetic switches, but otherwise the described pathways involved different genes. In both cases, a molecular pathway involving a small number of genes was described.

The comprehensive analysis presented here suggests that this feedback loop is much more extensive, involving numerous transcription factors that control a large and common set of genes. These genes not only include those induced during the cellular transformation process, but also many genes that are constitutively expressed yet are affected in a common fashion by these factors. In this regard, the set of genes induced by a given environmental stress is not well conserved across yeast species, whereas the overall category of genes is highly conserved[48]. By analogy with stable developmental states, we suggest that the critical transcription factors form a stable regulatory loop for each other's expression, thereby leading to a common set of target genes. In this view, transient induction of Src leads to changes in transcription factor activity or levels, and the altered state of transcription factors is self-reinforcing, leading to a new and stable state of gene expression.

Cancer occurs primarily as a consequence of somatic mutations and DNA methylation of tumor suppressor genes, and every cancer is genetically and epigenetically distinct. As such, an epigenetically stable cancer state is presumably not derived from evolutionary selection, but rather reflects a natural state of the wild-type organism. The simplest view is that this natural state represents a de-differentiated state in early development, where rapid growth is important. Thus, we suggest that the regulatory loop that is critical to maintain the stable transformed state in our model is not generated de novo, but rather reflects the induction

of a natural de-differentiated, rapid-growth state by a transient inflammatory stimulus.

## Methods

**Cell culture and cell transformation assays.** MCF-10A-ER-Src cells were cultured in DMEM/F12 medium with the supplements as previously described[22,23]. Tamoxifen (TAM, Sigma, H7904) 0.4 mM for 24 h was used to transform this inducible cell line when the cells were grown to 30% confluence. The transformation assay that measures growth under low attachment conditions has been described previously[36].

**Chromatin immunoprecipitation sequence.** ChIP was performed as described previously[33] with some modifications. Cells were treated with ethanol or tamoxifen (1 μM) for 24 h and then cross-linked using ethylene glycol bis (succinimidyl succinate) (EGS), disuccinimidyl glutarate (DSG) and disuccinimidyl suberate (DSS) mixture (2.5 μM each) for 45 min at room temperature. After this initial crosslinking, cells were further fixed using 1% formaldehyde for 20 min at room temperature and then quenched by glycine (0.125 M). Chromatin in sonication buffer (50 mM HEPES, pH7.5, 140 mM NaCl, 2 mM EDTA, 2 mM EGTA, 1% Triton-100, and 0.4% SDS) was sheared using Branson Microtip Sonifier 450 (12 cycles of 15 s at a sonication setting of output 4 and duty cycle 60%) to a size mostly between 100–150 bp. The sonicated chromatin solution was diluted to 0.085% SDS and immunoprecipitated with antibodies against H3K4me3 (ab8580), H3K27ac (ab4729), H3K4me1 (ab8895), H3K36me3 (ab9050), H3K27me3 (ab6002), and H3K9me3 (ab8898). Immunoprecipitated chromatin was decrosslinked using RNase Cocktail (Ambion, AM2286) and Pronase (Roche, 10165921001). ChIP DNA was end repaired, addition of "A" and adapters ligation and PCR amplification to produce ChIP-seq libraries. The DNA concentration was measured by Bioanalyzer before sequencing using Hiseq 2000 at the Bauer Core Facility, Harvard.

**DNase-seq.** The procedures for DNase treatment of chromatin and library preparation have been described previously[10].

**siRNA transfection and qPCR.** Cells were seeded for 24 h and then transfected with siRNAs, 50 nM (Dharmacon) and Lipofectamine RNAiMax (Life Technologies). siRNA sequences were in Supplementary Data 3. After 24 h, cells were split and treated with either ethanol or Tamoxifen (0.4 μM, Sigma-Aldrich, H7904) plus AZD0530 (0.4 nM, Selleck Chemicals, S1006) for 24 h. Total RNA was isolated using mRNeasy Mini Kit (Qiagen, No. 217004). Two microgram RNA was used for SYBR Green based. Primers were listed in Supplementary Data 4.

**RNA-seq library preparation.** Briefly, RNA was extracted using mRNeasy Mini Kit following the manufacturer's instruction. RNA-seq libraries were prepared using TruSeq Ribo Profile Mammalian Kit (Illumina, RPHMR12126) as per manufacturer's instruction. RNA-seq libraries were sequenced by Bauer Core Facility using Hiseq 2000.

**Analyses of time-series of mRNA expression data.** We profiled mRNA expression profiles using Affymetrix Human U133 2.0 A expression arrays, at 0 h, 1 h, 2 h, 4 h, 8 h, 12 h, 16 h, and 24 h upon *Src* oncogene induction with two biological replicates at each time point (GSE17941)[22]. The gene expression values were calculated by the RMA approach using Affymetrix Expression Console Software. We used MAS5 algorithm to estimate whether a gene is expressed in a sample and required the genes should be expressed in two biological replicates. Differently expressed genes were selected using the cutoff >1.5 fold change consistently in two biological replicates as compared to control in at least one time point during cell transformation. To test the validity of the cutoff, we randomized the fold-change values across genes 100,000 times, and applied the cutoff to estimate the false discovery rate (FDR) for differentially expressed genes as $<7 \times 10^{-3}$. The expression values were then mean-normalized and standardized. We used *K*-mean clustering to group differentially expressed genes into four coherent clusters, with median Pearson Correlation values >0.7 of genes in each cluster.

**DNase-seq and ChIP-seq data analyses.** Raw Fastq reads were aligned to human reference genome (hg19) using Bowtie[49] allowing up to 2 mismatches. Only the uniquely mappable reads were used for subsequent analyses. For DNase-seq data, we used MACS[50] to call peaks with the cutoff *P*-value $< 10^{-11}$ in at least one sample and using the following parameters "macs2 callpeak --llocal 1000000 -g 2.7e9". For ChIP-seq data for H3K27ac, H3K4me3, and H3K4me1, we used MACS[50] to call peaks with the cutoff *P*-value $< 10^{-8}$ in at least one sample and using the following parameters "macs2 callpeak --llocal 1000000 -g 2.7e9". For ChIP-seq for H3K27me3, H3K9me3 and H3K36me3, we used SICER[51] to call peaks with the cutoff *E*-value >40, window size 200 bp and gap size 600 bp, which is better for identifying broad read peaks. Then for each data type, we merged overlapping significant peaks from samples in different time points. For each

merged peak, its expression level in a sample was measured by the Reads per Million (RPM) value.

**Bayesian network model to identify potential functional TFBS.** To identify potential functional TFBS, we considered transcription factors with annotated Position weight matrix (PWM) in human, mouse and rate defined by MotifDB[35]. Based on those PWM, we used FIMO[52] to search potential TFBS in human genome (hg19) with default cutoff *E*-value $< 10^{-4}$.

For each potential TFBS *i* in the genome, its binding status is a hidden variable, either bound ($b_i = 1$) or unbound ($b_i = 0$). To estimate the binding probability, we hypothesized that a motif is more likely to be functional and bound by the factor, if it is closer to a transcription start site (TSS), show higher conservation during evolution, is more similar to the consensus sequence (higher PWM score) and is located more accessible chromatin regions. For each potential TFBS, we calculated distance to the closest TSS defined by refSeq (TSS_dist_i), normalized as $d_i = 1/(1 + $ TSS_dist_i/1000). The PWM score is calculated based on Fimo Score, as $f_i = $ (Fimo_Score_i-10)/10. We used the averaged PhastCons[53] score across 44 placental mammals to measure its conservation level as $c_i$. During our analyses, we found motifs located at edges of or close to DNase-seq peaks are less likely to be bound (as determined by ChIP-seq), and can cause false positives in prediction. So we calculated the number of DNase-seq reads 200 bp upstream and downstream a motif, respectively, and picked the lower number to represent the chromatin accessibility. We used the Bayesian Network Model (Table 1) to estimate the contribution of PWM score ($f_i$), distance to TSS ($d_i$), conservation level ($c_i$), and DNase I tag ($n_i$) to the probability of motif binding ($P(b_i = 1)$), as shown in Fig. 3b.

The contribution of PWM score ($f_i$), distance to TSS ($d_i$) and conservation levels ($c_i$) to TF binding probability ($y_i = P(b_i = 1)$) is modeled by logistic regression:

$$\log\left(\frac{y_i}{1 - y_i}\right) = \beta_0 + \beta_1 \times f_i + \beta_2 \times d_i + \beta_3 \times c_i$$

The TF binding probability ($y_i = P(b_i=1)$) is correlated with the chromatin accessibility, which is measured as the number of DNase-seq tags around the motif ($n_i$). The distribution is modeled by the negative binomial distribution:

$$P(n_i|b_i = 0) = \text{Negative Binomial}(n_i|K_0, r_0)$$
$$= \frac{(n_i + K_0 - 1)!}{n_i!(K_0 - 1)!}(1 - r_0)^{K_0} r_0^{n_i}$$

$$P(n_i|b_i = 1) = \text{Negative Binomial}(n_i|K_1, r_1)$$
$$= \frac{(n_i + K_1 - 1)!}{n_i!(K_1 - 1)!}(1 - r_1)^{K_1} r_1^{n_i}$$

The expectation–maximization (EM) algorithm was used to find maximum likelihood estimates of parameters in the model, including $\beta_0, \beta_1, \beta_2, K_0, r_0, K_1, r_1$. We randomly picked 10,000 motifs for training to learn the parameters of AccessTF, and applied the parameters to predict the binding status of another randomly picked 10,000 motifs for testing to evaluate the algorithm performance. We set the prior binding status based on the number of DNase I tags around the motif. Motifs with top 5% number of tags were set as bound and others were set as unbound. We tried different prior probabilities and obtained similar predicted posterior binding probabilities after converge. We picked the selected one as it converges quickest to get the parameter of AccessTF on the training set and the prediction performed the best to predict the motif binding status on the testing set.

To examine the performance of the algorithm in predicting the motif status for AP1 and STAT in ER-Src cells, we analyzed ChIP-seq data for AP1 (FOS, JUN and JUNB) and STAT3 factors, respectively. We used MACS[50] to call ChIP-seq peaks using the following parameters "macs2 callpeak --llocal 1000000 -g 2.7e9", with the cutoff *P*-value $< 10^{-8}$. The motifs with defined ChIP-seq peaks were considered as true positive, and those not overlapping with ChIP-seq peaks were considered as true negative.

## Table 1 Bayesian Network Model Parameters

| | Normalized value | Type | Range |
|---|---|---|---|
| PWM score ($f_i$) | (Fimo_Score_i-10)/10 | Continuous | (0, 1) |
| Distance to TSS ($d_i$) | 1/(1+TSS_dist_i/1000) | Continuous | (0, 1) |
| Conservation ($c_i$) | PhastCon Score in placenta | Continuous | (0, 1) |
| TF binding ($b_i$) | Hidden | Binary | 0, 1 |
| DNase I tag ($n_i$) | DNase-seq read # in 200 bp upstream or downstream from the motif | Continuous | (0, +) |

Using the same analyses procedure, we applied AccessTF to predict transcription factor binding sites in K562 cells using DNase-seq data in ENCODE project[34]. ChIP-seq data for transcription factors in K562 cells were used to measure the algorithm performance.

**TFScore**. We rank-ordered the candidate factors based on 4 following criteria:

(1) Relative contribution of the motif to general increased chromatin accessibility during the cell transformation. If the motif occurrence is higher, the corresponding TF is more likely to be important. For each accessible motif identified in AccessTF, we calculated the sum of reads in 200 nt upstream and downstream, normalized it to total number of reads and obtained the read per million (RPM) value to represent its surrounding chromatin accessibility in a sample. For each PWM, we calculated the sum of fold changes at 6 h and 24 h after Tamoxifen treatment relative to 0 h to represent its contribution to increased chromatin accessibility, using the following scoring definition. The regulation at 6 h: +0 ($<= 600$), +1 (601–1000); +2 (1001–1500); +3 (1501–3000); +4 (3001–5000); +5 (>5001). The regulation at 24 h: +0 ($<=2000$), +1 (2001–4000); +2 (1001–1500); +3 (1501–3000); +4 (3001–5000); +5 (>5001).

(2) Relative enrichment of motifs in promoter/enhancer regions of differentially expressed gene clusters. A factor is more likely to be important, if the motif enrichment is higher. For each PWM, we assigned the accessible motifs to the nearest closest expressed gene with the distance between the motif and TSS smaller than 100 kb. We also associated the motif and gene if the distance is within 20 kb. We used the Fisher Exact test to check the enrichment of the motifs in differentially expressed gene clusters (Fig. 1b) as compared to expressed genes which do not show differential expression. The –log10 (P-value) was used to indicate the relative enrichment, using the following scoring definition: +0 ($< = 8$); +1 (8–11); +2 (11–14); +3 (14–17); +4 (17–20); +5 (>20).

(3) If a transcription factor is significantly up-regulated over 1.5 fold, we added scoring +15. An up-regulated factor is likely to be more important

(4) For transcription factors in the same family which have similar binding motifs, we picked a representative PWM and rank-ordered their relative importance based on their expression levels. A factor expressed at a higher levels if more likely to be important. Suppose the highest expression level of genes in a family is E, Following is the scoring definition: +5 (=E); +3 (E/2–E); +0 (E/4–E/2); −5 (E/6–E/4); −10 ($<=$ E/6).

The final TFScore is the sum of the above four criteria.

**RNA-seq data analyses**. Raw reads were aligned to GENCODE[41] defined transcripts and then human reference genome (hg19) using Tophat[49] allowing up to 2 mismatches. Only the uniquely mappable reads were used for subsequent gene expression analyses. Gene expression levels were calculated as transcripts per million (TPM) value.

**Gene ontology analyses**. The Database for Annotation, Visualization and Integrated Discovery (DAVID)[54] was used for gene ontology analyses.

**TCGA data analyses**. RNA-seq gene expression and genetic annotation data for 1182 breast cancer patients were downloaded from TCGA database[37]. We calculated Spearman's rank correlation coefficient values between gene pairs among transcription factors and target genes. We also randomly selected expressed genes and calculated Spearman's correlation as the background distribution. We grouped breast cancer patients based on their genetic subtypes as following: Triple Negative (ER−, PR−, and HER2−), Luminal A (ER+, PR+, and HER2−), Luminal B (ER+, PR+, and HER2+), and HER2+ (ER−, PR−, and HER2+). To calculate the relative expression of a set of transcription factors and target genes, we first median-normalized the log2 gene expression levels across patient samples. And then we took the median normalized values across genes in a gene set to indicate the relative expression level of the gene set in a sample.

**Data availability**. All sequencing data that support the findings of this study have been deposited in the National Center for Biotechnology Information Gene Expression Omnibus (GEO) and are accessible through the GEO series accession numbers GSE100259, GSE100255, GSE100257 and GSE100258. All computational codes are available from the authors upon request.

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

## Acknowledgements

We thank John Stamatoyannopoulos for generating the DNase-seq data and Chaolin Zhang (Columbia University) for generous sharing of the computational codes of the Bayesian Network model. This work was supported by a fellowship from the Postdoc Programme of the German Academic Exchange Service, DAAD, (A.J.), the NIH Ruth L. Kirschstein National Research Service Award for postdoctoral fellowship (C.C.), the Searle Leadership Fund in the Life Sciences from Northwestern University (Z.J.), and research grants CA 107486 (K.S.) and K99 CA 207865 (Z.J.) from the National Institutes of Health.

## Author contributions

Z.J. conceived and performed all the bioinformatic analyses, L.H. performed all the siRNA experiments and RNA-seq library preparation, L.H., A.R., A.J., and C.S.C. performed the ChIP-seq experiments, Z.J., A.R., and K.S. analyzed the experimental and bioinformatic analyses and wrote the paper, and K.S. conceived the experimental design.

## Additional information

**Competing interests:** Thea authors declare no competing interests.

