## [Peer Review File · Nature Communications]

Reviewers' comments:

Reviewer #1 (Remarks to the Author):

In the paper entitled "Genome-scale identification of transcription factors that mediate an inflammatory network during breast cellular transformation" Zhe Ji and colleagues use high-throughput experiments/sequencing and integrative computational techniques to characterize transcriptional regulators that drive transformation of a breast epithelial cell line into stably transformed cells. This manuscript reports transcription factors that are apparently novel for this biological process. Novel computational tools to analyze TF binding and to prioritise important transcriptional regulators are developed as part of this study. The key transcription factors identified from the data are experimentally verified to affect the cell transformation. The manuscript seems a straightforward research project (in a positive way), where findings of candidate transcription factors and regulatory mechanisms are identified from genome-scale data, and the biological importance of these findings is verified by siRNA silencing experiments. The experimental validations support the findings, but the computational biology methods raise several questions that I describe below.

1. Use of statistics can be improved in several parts in the manuscript; otherwise it is difficult for a reader to get a overall (objective) understanding of the significance of the presented results. Identification of significant hits from transcriptome/genome/pathway -wide experiments inherently leads to statistical testing that involves possibly extensive multiple testing. While fixed genome-wide significance levels are commonly used and therefore satisfactory for peak detection from ChIP-seq (here $p < 10^{-11}$) and DNase-seq ($p < 10^{-8}$) data, the multiple testing issue should be taken into account in identification of differentially expressed genes and enriched pathways. Similarly, statistical testing and multiple testing correction would greatly strengthen the experimental siRNA validation at the end (Fig. 5)

2. For gene expression analysis, authors note that more modern analysis methods are used compared to an earlier work that used the same data. While microarray data preprocessing using the RMA tool is considered to be a standard, identification of differentially expressed genes using a fold-change analysis does not provide evidence for the reliability of the results. Authors need to quantify the statistical significance of differential gene expression using statistical tools such as Limma or similar (paired design with two replicates), together with multiple testing issue.

3. Authors propose a Bayesian network (or generally a probabilistic graphical model) based method for transcription factor (TF) binding analysis. While the method seems to perform reasonably well on validation data (ChIP-seq) this part of the manuscript is poorly described and has also other shortcomings. First and foremost, the proposed method is described only at qualitative level by describing variables/features it uses and showing the conditional (in)dependencies in Fig. 3B. Authors need to provide a detailed description of their method, starting with the underlying statistical model (that includes continuous, binary and discrete variables), its parameters, etc. For example, authors note that the DNase-seq read data is modeled as negative binomial distributed variables but that data type is listed as continuous-valued in a table on pages 19-20. Some details of the EM algorithm are also needed to verify/understand the expectation-maximization algorithm is applied correctly.

Authors demonstrate the accuracy of their TF binding analysis method using ChIP-seq from two cell lines: authors have measured AP-1 and Stat3 binding, and K562 data is taken from the ENCODE project. Authors need to describe in detail how they construct the positive (true binding) and negative (non-binding) sites for ROC/AUC analysis. A number of computational methods that use motif and chromatin accessibility information have been proposed to predict TF binding. While

the results (e.g. AUC-scores) look reasonably good, authors need to compare their method to existing state-of-the-art methods using a common data set. Authors emphasize that their method works on low coverage data (total sequencing reads per experiments are not reported anywhere) but existing tools can also tolerate low coverage.

4. DNase-seq and ChIP-seq analysis. Describe the details of MACS analysis for DNase-seq data. It is unclear if authors have implemented an analysis that is particularly meant for detecting enriched cutting loci. Concerning ChIP-seq data analysis, is a control used (input DNA, IgG, etc.)? Or is a non-uniform background taken into account otherwise?

Reviewer #2 (Remarks to the Author):

This is a data-rich study aiming to comprehensively map the transcriptional regulatory events underlying a model of inducible cellular transformation. The authors use RNA-seq, DHS and histone mark profiling, in a time course design, to identify over 40 transcription factors (TFs) that are likely to be involved in the regulatory program. They then experimentally test 20 of these predicted TFs, and find that transformation efficiency is indeed significantly altered upon knock-down of most of these TFs. They then select six of these confirmed TFs for further investigation, and generate RNA-seq profiles in knock-down conditions. This leads to identification of the genes that these TFs might target as part of their role in the regulatory cascade being explored.

The study is significant because of the catalog of information it presents, pertinent to the induced cellular transformation of breast epithelial cells. It represents a major advance in the extent of characterization of this system at the regulatory level. The paper is well written and is easy to read, and presents many useful and insightful observations as it reports various aspects of the data analysis. It is also methodologically important in so far as it confirms, and demonstrates convincingly in a new system, that a regulatory system can be characterized in great detail by mostly relying on expression and accessibility data, along with TF binding preferences (PWMs); this general approach has been used in previous studies as well, but in different contexts (e.g., Blatti et al. PMID 25791631, Duren et al PMID 28576882).

The computational methodology of the work, though well explained, comes out as rather underwhelming. For instance, one of the two major contributions – the Bayes net model to score binding sites – tackles a task that is very reminiscent of the CENTIPEDE algorithm, though no clear justification is provided for developing a new method and no comparisons are provided either. Prediction of TF binding, to the extent that these predictions can proxy for TF ChIP experiments, have also been tackled by several other studies and methods, e.g., gkm-SVM, and the recent DREAM challenge had many groups contributing their ideas and methods for this task. Nothing from this substantial prior literature is discussed, acknowledged, or compared to. As such, this part of the study looks oddly unjustified, which is unfortunate, because the overall study design could probably have been carried out with existing methods for this step. I do not have any great suggestion to offer the authors here, since they have already done the work using their new method, and I do not think it would be productive to change that. At this point, simply acknowledging the relevant literature on the topic would suffice.

The relative weakness of the computational methodology also comes across in the second main computational aspect – the TFScore method for ranking TFs. The method is a poorly explained combination of heuristics (e.g., “If a transcription factor is significantly up-regulated over 1.5 fold, we added scoring +15”, “Suppose the highest expression level of genes in a family is E, Following is the scoring definition: +5 (=E); +3 (E/2-E); +0 (E/4-E/2); -5 (E/6-E/4); -10 (<=E/6).”, “The final TFScore is the sum of the above four criteria”). The basic idea of scoring TFs for processes based on motif presence as well as their expression has been explored previously (e.g., Duren et

al PMID 28576882). I am not sure what novel idea the TFscore method incorporates, perhaps criterion (1) of including motif presence in differential accessibility peaks is a novel component. Once again, since the predictions made by this new method have already been tested and found to be quite well supported by these experiments, I do not think the authors need to or can change their method. Perhaps some clarification on the choice of their scoring scheme might make the method more appealing to others wanting to use it.

Specific comments:

The Methods section should clarify what were the types of motif sources used in the MotifDB database. If ChIP-seq data sets were used to learn motifs, as they often are, then the cross-validation of the Bayesian Network model is not a true cross-validation.

The evaluation of the Bayesian Network model for scoring motifs suggests that it mostly reflects the accessibility of the motifs. Where is the TF-specificity of the predictions coming from, if not from the TF motif strength? I suppose it comes from the first step, that of FIMO scans, which is done even before the BN scoring method is invoked. I request that the authors include in Figure 3C (and S4C) the performance of a baseline method that only used accessibility as a filter (and no Bayes net) on top of FIMO scans.

The last subsection of Results had an opportunity to examine the extent to which motif-based predictions of TF-gene relationships (part of the input to TFScore) agree with expression data under TF knock-down. In particular, one could have found out what fraction of the TF's predicted target genes are indeed differentially expressed upon knock-down of the TF, and how this compares to appropriate baselines/controls. The authors seem to have provided some information on this point, but in the opposite direction ("These enhancers and promoters typically lack one or more motifs, and hence direct binding sites, for transcription factors that nevertheless influence transcription of the gene"), where their finding is explainable using indirect influence or indirect binding. It would help to also use this opportunity to assess the precision of motif-based target predictions.

The authors note on page 20: "We tried different prior probabilities and obtained similar predicted posterior binding probabilities after converge. We picked the selected one as it converges quickest and the prediction performed the best." Picking a method because it performs best in predictions sounds like it was a violation of the training/testing exclusivity in a cross-validation setting. The performance values reported in Results are assumed to be cross-validation, so this sounds odd. This should be clarified.

Reviewer #3 (Remarks to the Author):

Ji et al. submitted to Nature Communications

In their paper Ji et al present an experimental and computational analysis of a model system of breast epithelial cell transformation. They employ the cell line MCF-10A, in which Src is overexpressed, as previously been described (REF 18). Oncogenic transformation is initiated by tamoxifen-induced overexpression of Src. Upon Src induction a time course analysis of chromatin conformation was combined with previously derived gene expression data. The authors generate a tool for the prediction of TF binding by combining data on chromatin accessibility, PWM score, distance to TSS and conservation to generate a predictor of TF binding. They then compare their predictive scores with scores obtained from ChIP-seq data for Stat3 and JunB obtained using the same cell line. In addition, the predictive scores are tested on a number of TFs, using ENCODE data for the K562 cell line. The authors show very high predictive scores for TF binding.

They next derive a score that measures the likelihood of a TF being involved in the transformation process, based on whether or not they bind in regulatory elements of differentially expressed genes, in regions with increased chromatin accessibility, and whether the factor or a close relative is upregulated in the transformation process. A potential list of TFs is arrived at and a number are tested and validated via siRNA-mediated downregulation experiments, with six of TFs profiled at the RNAseq level. They observe that these TFs share many targets.

The tool is potentially very useful for many different biologic data sets where accessibility data exists. Of the TFs the authors identify, a number are validated giving some confidence in the method.

Points to address:

Clearly the authors wish to develop a model that is relevant for breast cancer. They MCF-10A model they choose represents a model for ER- cancer. There is already a lot of data on gene expression patterns in ER- breast cancer (TCGA, METABRIC). For which of the TFs the authors identify as relevant in transformation is there evidence for a functional role in disease? Also regulatory networks in ER- breast cancer has been studied in a number of different systems. How do the results obtained relate to this previous work?

The last section of the paper, where common responses after siRNA downregulation of relevant TFs is assessed, is quite hard to follow the details. This is partially due to extremely short figure legend (Also see below.) For example Figure legend 6c only states "Dynamic regulation of gene expression in (A) during transformation. Without additional detail the analysis represented by the figure is not easily accessible. Furthermore Figure 6d "The gene transcription network mediated the TFs. The edges represent direct factor bind [should be: binding] sites in promoters/enhancers of targeted genes." Again this is not informative. The figure shows 30 genes. What were the criteria for choosing these 30 genes? When common up and downregulated genes are discussed, the authors mention 1326 genes, but little information is given how this is whittled down to 30. In general the authors should give more explanation in the figures. Another example includes figure 4, where no mention of blue text is made. It's possible to work out that TFs shown in blue were validated, but it would be helpful to state this. Can the authors please check all Figure legends for completeness.

My last criticism relates to the conclusion that an "inflammatory network" is identified, as claimed in the title. Indeed, Stat3 and NFkB are frequently involved in immune responses, however, when the targets of multiple of the identified TFs are examined by gene ontology, very general cellular processes are identified (cell cycle, translation, RNA processing, cell division, macromolecular complex organisation), that could conceivably be linked to most cellular functions. The authors do not present any statistically tested analysis that links the identified TFs to inflammation over other processes. To justify the title such an analysis needs to be carried out.

Minor points:

Supplementary Figure legend 4: There seem to be two different figure legends.

Reviewer 1

1. Use of statistics can be improved in several parts in the manuscript; otherwise it is difficult for a reader to get an overall (objective) understanding of the significance of the presented results. Identification of significant hits from transcriptome/genome/pathway -wide experiments inherently leads to statistical testing that involves possibly extensive multiple testing. While fixed genome-wide significance levels are commonly used and therefore satisfactory for peak detection from ChIP-seq (here $p < 10^{-11}$) and DNase-seq ($p < 10^{-8}$) data, the multiple testing issue should be taken into account in identification of differentially expressed genes and enriched pathways. Similarly, statistical testing and multiple testing correction would greatly strengthen the experimental siRNA validation at the end (Fig. 5)

Response: For all relevant cases, we now use Benjamini P -values for GO analyses to address the multiple testing issue (this was only done for some situations in the original manuscript). We use Fisher's Exact test to support the conclusion of siRNA validation experiment, with a P -value $< 5 \times 10^{-3}$, comparing validation rates for transcription factors with high TFScores versus low ones.

2. For gene expression analysis, authors note that more modern analysis methods are used compared to an earlier work that used the same data. While microarray data preprocessing using the RMA tool is

considered to be a standard, identification of differentially expressed genes using a fold-change analysis does not provide evidence for the reliability of the results. Authors need to quantify the statistical significance of differential gene expression using statistical tools such as Limma or similar (paired design with two replicates), together with multiple testing issue.

Response: Due to the inducible nature of the cell transformation model and multiple time points, the fold-change values can be slightly variable across biological replicates. To deal with this issue, we selected significant differentially expressed genes using the cutoff >1.5 -fold change consistently in two biological replicates as compared to control in at least one time-point during cell transformation, and we did not require low standard deviation between replicates. To estimate the confidence level of the cutoff, we generated the background distribution by randomizing the gene expression fold change values across genes in samples 100,000 times, and applied the cutoff to estimate the false discovery rate (FDR). The cutoff we use to select significant differentially expressed genes has the $FDR < 7 \times 10^{-3}$.

3. Authors propose a Bayesian network (or generally a probabilistic graphical model) based method for transcription factor (TF) binding analysis. While the method seems to perform reasonably well on validation data (ChIP-seq) this part of the manuscript is poorly described and has also other shortcomings. First and foremost, the proposed method is described only at qualitative level by describing variables/features it uses and showing the conditional (in)dependencies in Fig. 3B. Authors need to provide a detailed description of their method, starting with the underlying statistical model (that includes continuous, binary and discrete variables), its parameters, etc. For example, authors note that the DNase-seq read data is modeled as negative binomial distributed variables but that data type is listed as continuous-valued in a table on pages 19-20. Some details of the EM algorithm are also needed to verify/understand the expectation-maximization algorithm is applied correctly.

Authors demonstrate the accuracy of their TF binding analysis method using ChIP-seq from two cell lines: authors have measured AP-1 and Stat3 binding, and K562 data is taken from the ENCODE project. Authors need to describe in detail how they construct the positive (true binding) and negative (non-binding) sites for ROC/AUC analysis. A number of computational methods that use motif and chromatin accessibility information have been proposed to predict TF binding. While the results (e.g. AUC-scores) look reasonably good, authors need to compare their method to existing state-of-the-art methods using a common data set. Authors emphasize that their method works on low coverage data (total sequencing reads per experiments are not reported anywhere) but existing tools can also tolerate low coverage.

Response: We now add detailed description of the Bayesian network, including the parameters and mathematical formula. To construct the positive and negative sites, we used MACS to call ChIP-seq peaks for transcription factors using the following parameters “macs2 callpeak --local 1000000 -g 2.7e9”, with the cutoff P -value $< 10^{-8}$. The motifs with defined ChIP-seq peaks were considered as true positive, and those not overlapping with ChIP-seq peaks were considered as true negative. The revised manuscript also contains additional citations to previous methods, as suggested.

Comparisons of AccessTF to DNase I footprinting methods: Although AccessTF and DNase I footprinting algorithms both utilize DNase I-seq data, they measure different molecular entities so direct comparisons are apples:oranges and not meaningful. In particular, AccessTF utilizes DNase I hypersensitive regions (typically a few hundred base pairs, and virtually every sequence read corresponds to a hypersensitive region; genome-scale identification of such regions can be done with ~ 30 million reads. AccessTF doesn't directly identify TF binding sites, but rather predicts TF binding probability based on DNase I hypersensitivity and other parameters. Our AUC analysis indicates AccessTF to be highly predictive.

In contrast, the computational methods mentioned by Reviewer 1 are designed to detect DNase footprints, a direct measure of protein binding. DNase I footprints are small protected regions (~20 bp) typically within the larger hypersensitive regions utilized by AccessTF. DNase I footprints can only be identified by obtaining a vast number of sequencing reads with the hypersensitive regions in order to see the small regions that are protected. The published methods for detecting DNase I footprints perform well on ENCODE data, with >500 million reads/dataset, but they cannot handle low number of reads that are sufficient to map DNase I hypersensitive regions. We tested several state-of-the-art methods using our ER-Src data. The older version CENTIPEDE resulted in AUC values of ROC curve are ~0.75 for AP1 and STAT, while AccessTF results in AUC values > 0.95. For the recent version msCENTIPEDE, the algorithm could not converge in 3 days for AP1 and STAT motifs using our data (40 - 70 million reads).

The HINT algorithm, one of the best current algorithms, tolerated low read numbers, and outputted the predicted footprints. We compared the performance of HINT vs. AccessTF by randomly sampling 10-70 million reads in ER-Src cells, and used F1 score to measure the precision and recall of the algorithms. AccessTF consistently performs much better than HINT (mean and SD from 5 randomly sampled datasets; figure below).

In summary, AccessTF and DNase footprinting algorithms use DNase-seq data and are designed to identify TF target sites *in vivo*. However, AccessTF infers these sites from hypersensitive regions and other parameters, whereas DNase footprints provide direct evidence for binding. For the ultimate goal of identifying *in vivo* target sites, both methods are quite accurate, and they have advantages and disadvantages. The main value of AccessTF is that one needs ~10-fold less sequencing depth, which is a major advantage when analyzing multiple samples as is the case here. But, these methods are different, so direct comparisons of the type suggested aren't meaningful.

Comparisons of AccessTF to chromatin accessibility methods: AccessTF is more similar to earlier methods that search for TF motifs in DNase I hypersensitive regions. However, as discussed in the text, these methods are clearly inferior, which is why we developed AccessTF. Unlike earlier methods, AccessTF accounts for where the motif is located within the accessible region, it accounts for the level of chromatin accessibility, it utilizes other parameters such as conservation, and it yields binding

probabilities for each motif. Furthermore, the logic of AccessTF is the opposite of the other methods in that it starts from all motifs for all factors in the human genome and then determines their binding probability.

4. DNase-seq and ChIP-seq analysis. Describe the details of MACS analysis for DNase-seq data. It is unclear if authors have implemented an analysis that is particularly meant for detecting enriched cutting loci. Concerning ChIP-seq data analysis, is a control used (input DNA, IgG, etc.)? Or is a non-uniform background taken into account otherwise?

Response: We added parameters for MACS in the method section. We used 1,000,000 nt genomic region around as the background distribution.

Reviewer 2

1. This is a data-rich study aiming to comprehensively map the transcriptional regulatory events underlying a model of inducible cellular transformation. The authors use RNA-seq, DHS and histone mark profiling, in a time course design, to identify over 40 transcription factors (TFs) that are likely to be involved in the regulatory program. They then experimentally test 20 of these predicted TFs, and find that transformation efficiency is indeed significantly altered upon knock-down of most of these TFs. They then select six of these confirmed TFs for further investigation, and generate RNA-seq profiles in knock-down conditions. This leads to identification of the genes that these TFs might target as part of their role in the regulatory cascade being explored.

The study is significant because of the catalog of information it presents, pertinent to the induced cellular transformation of breast epithelial cells. It represents a major advance in the extent of characterization of this system at the regulatory level. The paper is well written and is easy to read, and presents many useful and insightful observations as it reports various aspects of the data analysis. It is also methodologically important in so far as it confirms, and demonstrates convincingly in a new system, that a regulatory system can be characterized in great detail by mostly relying on expression and accessibility data, along with TF binding preferences (PWMs); this general approach has been used in previous studies as well, but in different contexts (e.g., Blatti et al. PMID 25791631, Duren et al PMID 28576882).

Response: As requested, we cite and discuss more papers that involve integrative analyses of chromatin accessibility and transcriptional profiles to identify transcription factors.

2. The computational methodology of the work, though well explained, comes out as rather underwhelming. For instance, one of the two major contributions – the Bayes net model to score binding sites – tackles a task that is very reminiscent of the CENTIPEDE algorithm, though no clear justification is provided for developing a new method and no comparisons are provided either. Prediction of TF binding, to the extent that these predictions can proxy for TF ChIP experiments, have also been tackled by several other studies and methods, e.g., gkm-SVM, and the recent DREAM challenge had many groups contributing their ideas and methods for this task. Nothing from this substantial prior literature is discussed, acknowledged, or compared to. As such, this part of the study looks oddly unjustified, which is unfortunate, because the overall study design could probably have been carried out with existing methods for this step. I do not have any great suggestion to offer the authors here, since they have already done the work using their new method, and I do not think it would be productive to change that. At this point, simply acknowledging the relevant literature on the topic would suffice.

Response: See section above for our detailed response to point 3 of Reviewer 1.

3. The relative weakness of the computational methodology also comes across in the second main computational aspect – the TFscore method for ranking TFs. The method is a poorly explained combination of heuristics (e.g., “If a transcription factor is significantly up-regulated over 1.5 fold, we added scoring +15”, “Suppose the highest expression level of genes in a family is E, Following is the scoring definition: +5 (=E); +3 (E/2-E); +0 (E/4-E/2); -5 (E/6-E/4); -10 (<=E/6).”, “The final TFscore is the sum of the above four criteria”). The basic idea of scoring TFs for processes based on motif presence as well as their expression has been explored previously (e.g., Duren et al PMID 28576882). I am not sure what novel idea the TFscore method incorporates, perhaps criterion (1) of including motif presence

in differential accessibility peaks is a novel component. Once again, since the predictions made by this new method have already been tested and found to be quite well supported by these experiments, I do not think the authors need to or can change their method. Perhaps some clarification on the choice of their scoring scheme might make the method more appealing to others wanting to use it.

Response: This method utilizes 4 criteria to predict the functional importance of a TF. Previous papers only considered 1 or 2 criteria listed above. As shown in Figure 4b, the predicted functional TFs do not necessarily meet each of the 4 criteria. Integrating 4 parameters provides a more powerful prediction. We added more description of the rationales in the Results and Methods sections. It should also be noted that previous papers do not comprehensively evaluate TFs for their functional importance in a given system, and, our paper provides experimental validation for TFScore.

4. The Methods section should clarify what were the types of motif sources used in the MotifDB database. If ChIP-seq data sets were used to learn motifs, as they often are, then the cross-validation of the Bayesian Network model is not a true cross-validation.

Response: We did not use ChIP-seq data to learn motifs; instead we used well-characterized and high quality PWMs in mammals (human, mouse and rat) for transcription factors defined in the MotifDB database. We used Fimo to search genome-wide motifs for each factor, selected high quality motif with E-value of $<1E-4$, and then used AccessTF to predict the binding probability for each motif. ChIP-seq data for transcription factors were only used to test the performance of the algorithm. We now added more detailed description in the Methods section.

5. The evaluation of the Bayesian Network model for scoring motifs suggests that it mostly reflects the accessibility of the motifs. Where is the TF-specificity of the predictions coming from, if not from the TF motif strength? I suppose it comes from the first step, that of FIMO scans, which is done even before the BN scoring method is invoked. I request that the authors include in Figure 3C (and S4C) the performance of a baseline method that only used accessibility as a filter (and no Bayes net) on top of FIMO scans.

Response: We don't understand the request to include in Figure 3C (and S4C) the performance of a baseline method that only used accessibility as a filter (and no Bayes net) on top of FIMO scans. We used FIMO to search potential binding motifs in the hg19 genome and did not limit our search to open chromatin regions. It is true that accessibility is the most important parameter for AccessTF (shown in paper), but the degree of accessibility and the location of the motif within the accessible region are also important. And more importantly, we obtain a binding probability for each motif.

6. The last subsection of Results had an opportunity to examine the extent to which motif-based predictions of TF-gene relationships (part of the input to TFScore) agree with expression data under TF knock-down. In particular, one could have found out what fraction of the TF's predicted target genes are indeed differentially expressed upon knock-down of the TF, and how this compares to appropriate baselines/controls. The authors seem to have provided some information on this point, but in the opposite direction ("These enhancers and promoters typically lack one or more motifs, and hence direct binding sites, for transcription factors that nevertheless influence transcription of the gene"), where there finding is explainable using indirect influence or indirect binding. It would help to also use this opportunity to assess the precision of motif-based target predictions.

Response: We agree and now add a sentence in the Discussion to say that transcription factors can be recruited to "non-cognate" motifs via protein-protein interactions, and these will be missed. ChIP-seq doesn't have this problem, but ChIP-seq can only be performed for a limited number of factors, certainly not a global analysis. The motif-based analysis here provides very good estimation of factor

binding through motif recruitment, and a powerful prediction of functional transcription factors in a cost-effective manner. Of course, ChIP-seq and other approaches are critical to elucidate mechanisms in more detail. Importantly, the limitation of motif-based predictions does not affect the conclusions of the manuscript.

7. The authors note on page 20: “We tried different prior probabilities and obtained similar predicted posterior binding probabilities after converge. We picked the selected one as it converges quickest and the prediction performed the best.” Picking a method because it performs best in predictions sounds like it was a violation of the training/testing exclusivity in a cross-validation setting. The performance values reported in Results are assumed to be cross-validation, so this sounds odd. This should be clarified.

Response: It is a common practice to pick the best parameters in a computational model to achieve the best performance. We used ChIP-seq binding sites of transcription factors to get the positive/negative sites of motifs to test the performance of our algorithm, and the predictions were solely based on DNase-seq data. There is no violation of the training/testing to optimize the input parameters of AccessTF, because the set of motifs in the training set (randomly picked 10,000 motifs) were different from those used for testing (another randomly picked 10,000 motifs).

Reviewer 3

1. Clearly the authors wish to develop a model that is relevant for breast cancer. They MCF-10A model they choose represents a model for ER- cancer. There is already a lot of data on gene expression patterns in ER- breast cancer (TCGA, METABRIC). For which of the TFs the authors identify as relevant in transformation is there evidence for a functional role in disease? Also regulatory networks in ER- breast cancer has been studied in a number of different systems. How do the results obtained relate to this previous work?

Response: The purpose of this paper is not “to develop a model that is relevant for ER-negative breast cancer”. As stated in the initial section and the Introduction, we have used this model for 10 years, and published many papers demonstrating relevance to human cancer. Moreover, those analyses indicate that the model is relevant for multiple cancer types, not just ER-negative breast cancer, and this is hardly surprising. Instead, the purpose of this paper is to perform a genome-scale identification of TFs important for transformation in this model; we are unaware of anything like this in the literature. Nevertheless, in response to this comment, the revised manuscript directly analyzes the TFs identified here (new Fig. 7). Specifically, we now show that the TFScore predicted transcription factors and target genes show positive expression correlation among 1,182 breast cancer patient samples from TCGA database. And triple negative breast cancers tend to have higher expression levels of transcription factors and targets, as compared to ER+ breast cancers. We also note that several our identified factors such as CEBPB, HIF1a, ETS2, FHL2, TCF7L2 and NFE2L2, have known oncogenic roles.

2. The last section of the paper, where common responses after siRNA downregulation of relevant TFs is assessed, is quite hard to follow the details. This is partially due to extremely short figure legend (Also see below.) For example Figure legend 6c only states “Dynamic regulation of gene expression in (A) during transformation. Without additional detail the analysis represented by the figure is not easily accessible. Furthermore Figure 6d “The gene transcription network mediated the TFs. The edges represent direct factor bind [should be: binding] sites in promoters/enhancers of targeted genes.” Again this is not informative. The figure shows 30 genes. What were the criteria for choosing these 30 genes? When common up and downregulated genes are discussed, the authors mention 1326 genes, but little information is given how this is whittled down to 30. In general the authors should give more explanation in the figures. Another example includes figure 4, where no mention of blue text is made. It’s possible to

work out that TFs shown in blue were validated, but it would be helpful to state this. Can the authors please check all Figure legends for completeness.

Response: Regarding Fig. 6, we have added more description and have hopefully clarified the confusion. We have also checked (and in some cases modified) this and other Figure legends. The 30 genes were picked as representatives, nothing more.

3. My last criticism relates to the conclusion that an “inflammatory network” is identified, as claimed in the title. Indeed, Stat3 and NFkB are frequently involved in immune responses, however, when the targets of multiple of the identified TFs are examined by gene ontology, very general cellular processes are identified (cell cycle, translation, RNA processing, cell division, macromolecular complex organisation), that could conceivably be linked to most cellular functions. The authors do not present any statistically tested analysis that links the identified TFs to inflammation over other processes. To justify the title such an analysis needs to be carried out.

Response: The inflammatory aspect of our ER-Src model has been described in many of our papers (first in 2009). Again, the purpose of this paper is the genome-scale identification of important TFs, something not done previously. It is true that many of our identified TFs and downstream target genes are not directly involved in inflammation, yet nevertheless are important for transformation. This is not surprising, because most biological processes (e.g. developmental state) depend on specific and non-specific factors. The unquestioned inflammatory network that is critical for ER-Src transformation does not exclude the role of other transcription factors or processes (e.g. cell cycle). For genes showing consistent down-regulation upon 6 factor knockdowns, we compared inducible genes during the cell transformation, with genes in other enriched pathways. Inducible genes are more likely to be the direct bound target genes of 6 functional TFs we examined (4.6 AccessTF-predicted binding sites per gene vs. 3.2 sites per gene; P-value < 10^{-12} , Wilcoxon Rank Sum Test).

Minor point: We fixed the error of multiple Supplementary Figure 4 legends.

Reviewers' comments:

Reviewer #2 (Remarks to the Author):

The authors have addressed several of my comments from the previous review, except the following minor points.

Authors' response: "We did not use ChIP-seq data to learn motifs; instead we used well characterized and high quality PWMs in mammals (human, mouse and rat) for transcription factors defined in the MotifDB database"

Comment: MotifDB includes a number of PWMs derived from ChIP-seq peaks. Neither the revised manuscript nor the rebuttal clarifies that such motifs were avoided in reporting cross-validation results.

Authors' response: "We used ChIP-seq binding sites of transcription factors to get the positive/negative sites of motifs to test the performance of our algorithm, and the predictions were solely based on DNase-seq data. There is no violation of the training/testing to optimize the input parameters of AccessTF, because the set of motifs in the training set (randomly picked 10,000 motifs) were different from those used for testing (another randomly picked 10,000 motifs)."

Comment: When saying "We picked the selected one as it converges quickest and the prediction performed the best" (in text), if the authors mean performance on test data, then it is fine and the authors just need to add the phrase "on training data" at the end of this sentence. As stated, a reader can easily interpret this as meaning "performed the best on test data", which is the issue I was worried about.

Reviewer #3 (Remarks to the Author):

In this second response, I will primarily concentrate on points previously raised & addressed in the letter by the authors.

1. The first point asked to put the work into context of previously published work. The authors now provide Fig 7 with data to show how their identified genes behave in different tumour data sets. This is helpful and confirms their findings are most relevant for ER- and HER2 cancers, which tend to be the most proliferative type of BC. However, in other aspects this could be further improved. In the introduction the authors state: "However, a comprehensive analysis of cellular transformation in this or any other model of cellular transformation has not been described." and in the letter: "We are unaware of genome wide identification of TFs important for transformation". There are a number of studies looking at the genes - including TFs (although not exclusively TFs) - driving carcinogenesis and transformation. An example (there are others) can be found in Marcotte et al. (Cancer Discovery 2012, 2:172-189) in which the authors derive essentiality scores for genes by transfection shRNA libraries targeting 16,000 genes into 72 cell lines, including 29 breast cancer cell lines. It would be interesting to compare the results obtained in that study with the results obtained here, especially as the authors suggest that their approach is "loosely analogous to a genetic screen".

2. The figure legends are now clearer.

3. I continue to feel that the emphasis on inflammation in the title of the paper is misleading, as this paper (as opposed to previous work by the authors) does not present any evidence for an inflammatory involvement. Clearly this is an editorial decision.

The authors wish to carry out a “genome-scale identification of important TFs” and find that of the 11 most important TFs, 5 belong to the Fos/Jun family, with numbers 17 and 18 (ATF3 and BATF3) being Fos-related TFs that bind very similar sites. It seems to me that the overwhelming conclusion is that the AP-1 heterodimer & associated factors are the key drivers of the response. After that many different processes seem to contribute, including inflammation. However, up to point the choice of emphasis/presentation of the results is up to the authors.

Reviewer #4 (Remarks to the Author):

Authors have improved the statistics and the description of methods. This has settled most of the initial requests. I am, however, not yet convinced on the performance of methods for predictions of TF binding sites from open chromatin data. Moreover, I have new concerns about the final selection of TFs, which were verified with siRNA.

Major Aspects

1. Footprinting methods and AccessTF perform predictions on distinct genomic resolutions. While footprints are based on a binding site resolution (motif centric), AccessTF works on a peak resolution (peak centric). It is not difficult to conceive simple (or complex) approaches to make peak centric predictions from a motif centric method, i.e. select the maximum footprint score of a TF inside a peak. This would make both types of methods perfectly comparable. Indeed, authors were able to compare AccessTF with the footprinting method HINT. It is not clear why this analysis is based on F1 scores, while AUPR and AUC scores were used for other similar comparisons. F1 only evaluates precision and recall for a given number of positive calls. AUPR evaluates for all possible positive calls. Authors should provide AUPR curves/values of the comparison of HINT and AccessTF. Also, peak centric detection of binding site was the objective of the ENCODE Dream challenge, as noted by reviewer 2. These are based on more sophisticated features than AccessTF and should be acknowledged by the authors. Here are two examples:

<https://www.biorxiv.org/content/biorxiv/early/2017/12/06/230011.full.pdf>

<https://www.biorxiv.org/content/biorxiv/early/2017/06/18/151274.full.pdf>

Finally, these evaluations (AUPR of HINT) and its description needs to be integrated in the manuscript.

2. Authors do not describe how the 20 TFs were selected from the list of 50 top ranked TFs (blue factors in figure 4). Why were not the top ranked 20 TFs selected? Authors should complement their analysis with siRNA for other top scoring TFs (JUNB, IRF6, FOS, BCL6, PRDM1, R2F1). Also, they should include an evaluation of how the transformation efficiency of factors correlate with the TF score. The relative value of TF score should also be associated to transformation efficiency of the factors.

Minor Aspects

1. Authors should provide data used in the evaluation of AccessTF: coordinates of peaks and motifs used for definition of positives and negatives.

Reviewer 2

1. MotifDB includes a number of PWMs derived from ChIP-seq peaks. Neither the revised manuscript nor the rebuttal clarifies that such motifs were avoided in reporting cross-validation results

Response: The PWMs from either MotifDB or ChIP-seq data provide the information of consensus binding motifs of transcription factors. We first used a relatively stringent cutoff to select high quality factor binding sites across human genome with cutoff FIMO E-value $< 1E-4$. AccessTF predicts the binding status of the motifs mostly based on chromatin accessibility in the specific cell type. The value of AccessTF is to define cell-type specific functional binding motif located in open chromatin regions, and to filter out non-functional binding motifs ($>90\%$ out of total) located in close chromatin regions. As we showed in Figures 3E and S4C, the motif strength (similarity to PWM) provides minor contribution to the prediction, and the most important parameter is chromatin accessibility. As a result, there is no cross-validation issue in the prediction, even if the PWM is learned from ChIP-seq data.

2. When saying “We picked the selected one as it converges quickest and the prediction performed the best” (in text), if the authors mean performance on test data, then it is fine and the authors just need to add the phrase “on training data” at the end of this sentence. As stated, a reader can easily interpret this as meaning “performed the best on test data”, which is the issue I was worried about.

Response: AccessTF was performed in the manner expected by the Reviewer, and we have clarified the confusing sentence. AccessTF uses the expectation-maximization algorithm to estimate the parameters of the Bayesian Network Model on the training set (converging to the maximum likelihood). Then these parameters were used to estimate the motif binding status on an independent testing set. The prediction accuracy on the testing set was used to evaluate the overall algorithm performance. This is a common practice of machine learning to avoid overfitting. We now added more explanation, “*We picked the selected one as it converges quickest to get the parameters of the Bayesian Network Model based on the training set, and the prediction performed the best to predict the motif binding status on the testing set*”

Reviewer 3

1. The first point asked to put the work into context of previously published work. The authors now provide Fig 7 with data to show how their identified genes behave in different tumour data sets. This is helpful and confirms their findings are most relevant for ER- and HER2 cancers, which tend to be the most proliferative type of BC. However, in other aspects this could be further improved. In the introduction the authors state: "However, a comprehensive analysis of cellular transformation in this or any other model of cellular transformation has not been described." and in the letter: "We are unaware of genome wide identification of TFs important for transformation". There are a number of studies looking at the genes - including TFs (although not exclusively TFs) - driving carcinogenesis and transformation. An example (there are others) can be found in Marcotte et al. (Cancer Discovery 2012, 2:172-189) in which the authors derive essentiality scores for genes by transfection shRNA libraries targeting 16,000 genes into 72 cell lines, including 29 breast cancer cell lines. It would be interesting to compare the results obtained in that study with the results obtained here, especially as the authors suggest that their approach is "loosely analogous to a genetic screen".

Response: Our statement "comprehensive analysis...has not been described" specifically referred to identification of DNA-binding transcription factors, their target sites, and transcriptional circuitry. This was the goal and achievement of the work. The shRNA/CRISPR screening approach is certainly an excellent way to identify functionally important TFs, but it does not identify target sites or define transcriptional regulatory circuits. On the other hand, this genetic approach identifies non-TF genes important in the process. As such, these approaches are really designed for different purposes and are complementary. A direct comparison of TFs identified in our study vs. these genetic studies cannot be done, because the cell lines are different. However, in the revised paper, we do discuss comprehensive genetic screens as an alternative approach.

2. The inflammatory aspect of our transformation model is well established in our previous papers. I continue to feel that the emphasis on inflammation in the title of the paper is misleading, as this paper (as opposed to previous work by the authors) does not present any evidence for an inflammatory involvement. Clearly this is an editorial decision. The authors wish to carry out a "genome-scale identification of important TFs" and find that of the 11 most important TFs, 5 belong to the Fos/Jun family, with numbers 17 and 18 (ATF3 and BATF3) being Fos-related TFs that bind very similar sites. It seems to me that the overwhelming conclusion is that the AP-1 heterodimer & associated factors are the key drivers of the response. After that many different processes seem to contribute, including inflammation. However, up to point the choice of emphasis/presentation of the results is up to the authors.

Response: As noted by the Reviewer, the inflammatory aspect of our transformation model is well established in previous papers. Thus, our use of the term "inflammatory" in the title simply refers to what has been previously established, not a new finding in the paper. Results in the paper are consistent with this, as many of the functional transcription factors we identified are related to inflammation. We certainly agree that AP-1 factors are very important for transformation in our model (as are other factors), but we do not state or even mean to imply that inflammatory factors are more important than others.

Reviewer 4

1. Footprinting methods and AccessTF perform predictions on distinct genomic resolutions. While footprints are based on a binding site resolution (motif centric), AccessTF works on a peak resolution (peak centric). It is not difficult to conceive simple (or complex) approaches to make peak centric predictions from a motif centric method, i.e. select the maximum footprint score of a TF inside a peak. This would make both types of methods perfectly comparable. Indeed, authors were able to compare AccessTF with the footprinting method HINT. It is not clear why this analysis is based on F1 scores, while AUPR and AUC scores were used for other similar comparisons. F1 only evaluates precision and recall for a given number of positive calls. AUPR evaluates for all possible positive calls. Authors should provide AUPR curves/values of the comparison of HINT and AccessTF. Also, peak centric detection of binding site was the objective of the ENCODE Dream challenge, as noted by reviewer 2. These are based

on more sophisticated features than AccessTF and should be acknowledged by the authors. Here are two examples:

Response: This comment deals with a very minor issue that has nothing to the validity of our AccessTF method, which is established by direct experimentation and is not questioned by any of the other Reviewers. In addition, the comment doesn't relate to anything in the paper, but rather an issue raised in our original rebuttal letter involving comparisons of AccessTF to footprinting methods.

As stated in that original rebuttal and in agreement with Reviewer 4, AccessTF and footprinting methods work on different resolutions, and the basic measurements of DNase hypersensitivity and DNase footprints are different molecular entities; hence, one can only make apples:oranges comparisons. As stated in the original rebuttal, it is obvious on first principles of the assays themselves (i.e. independent of algorithms) that one needs many fewer sequencing reads for DNase hypersensitivity than for DNase footprints, and this is stated in the paper. Nevertheless, for the original rebuttal, we performed a straightforward comparison of AccessTF and other methods as they are published, and the result just confirmed the obvious. We presented F1 scores, because HINT only outputs the significant footprints it identifies and does not provide predicted *P*-values for each site for users. We do not have the data to estimate the AUPR values, so we used F1 scores to compare the performance of the algorithms using the cutoffs. It is not reasonable to ask us “conceive” a new approach to make this evaluation. Importantly, the relevant point mentioned in our paper was not to denigrate or compare other algorithms, but rather to illustrate the difference between footprinting and hypersensitivity assays. Reviewer 4 also provided 2 examples of papers we should acknowledge. However, neither of these papers are peer reviewed, and both of them were submitted to Biorxiv after the original submission or our paper (and one of them after our first resubmission). This is a very long answer to a trivial issue.

2. Authors do not describe how the 20 TFs were selected from the list of 50 top ranked TFs (blue factors in figure 4). Why were not the top ranked 20 TFs selected? Authors should complement their analysis with siRNA for other top scoring TFs (JUNB, IRF6, FOS, BCL6, PRDM1,R2F1). Also, they should include an evaluation of how the transformation efficiency of factors correlate with the TFScore. The relative value of TF score should also be associated to transformation efficiency of the factors.

Response: The 20 TFs were selected randomly from the top 50. This was an arbitrary choice, just like selecting the top 20 would be an arbitrary choice. In fact, the best and indeed standard ENCODE method for validation (I speak as a former member of ENCODE who was part of making the standards) is to sample throughout a rank ordered list to get an idea of where the cutoff is between positive vs. negative experimental outcomes. In fact, we didn't even reach this point, because such a high percentage of the randomly chosen factors within the top 50 were important for transformation (and also a few factors with low TFScores as negative controls). The suggestion to test additional factors within the top 20 is unnecessary, as it is virtually certain from the results in the top 50 that almost all will be functionally important. Most importantly, this comment is irrelevant to the central conclusion that a very high percentage of TFs predicted by TFScore to be important actually are.

Lastly, Fig. 5B suggests that transformation efficiency upon factor knockdown correlates poorly with TFScore. This is hardly surprising given that the observed transformation effect depends on numerous experimental factors that are not part of TFScore (e.g. degree of knockdown, functional redundancy of TFs that recognize the same sequence, % of wt protein level needed for a phenotypic effect, etc.).

REVIEWERS' COMMENTS:

Reviewer #2 (Remarks to the Author):

My concerns are now fully addressed. Thanks for the clarifications.

Reviewer #3 (Remarks to the Author):

The authors have addressed my previous criticisms.

I noticed one minor error:

page 3: ...and they ARE generated...(are is missing in the text)

Reviewer #4 (Remarks to the Author):

In their response letter, authors indicate that two of my comments are minor issues. Here are further clarifications on these issues, which are not minors in my opinion. These can be improved by modifications of the manuscript and inclusion of references.

Comments on Point 1

"Response: This comment deals with a very minor issue that has nothing to the validity of our AccessTF method, which is established by direct experimentation and is not questioned by any of the other Reviewers. In addition, the comment doesn't relate to anything in the paper, but rather an issue raised in our original rebuttal letter involving comparisons of AccessTF to footprinting methods."

This points were raised by me and another referee, as the manuscript includes several discussions on the fact DHS/AccessTF is able to perform TF binding prediction on libraries with less number of reads than footprint methods. Here is an example of such (page 14):

"As such, footprinting methods require much higher (~10 times more) sequencing depth and hence are considerably more expensive, especially for experiments involving multiple samples. Thus, while computational methods to detect DNase I footprints (38-40 perform well on DNase-seq data from ENCODE41 (> 500 million uniquely mappable reads for each dataset), they work poorly to identify genome-wide TF binding status from the data used here that contains only 40-70 million uniquely mappable reads per sample."

The current manuscript includes no single evidence of the poor performance of footprints on libraries with low number of reads or any evidence that AccessTF performs better than any method for prediction of binding sites. Authors need to remove such statements from the paper or provide experimental data supporting these.

Comments on Point 2

"The 20 TFs were selected randomly from the top 50. This was an arbitrary choice, just like selecting the top 20 would be an arbitrary choice. In fact, the best and indeed standard ENCODE method for validation (I speak as a former member of ENCODE who was part of making the standards) is to sample throughout a rank ordered list to get an idea of where the cutoff is between positive vs. negative experimental outcomes. In fact, we didn't even reach this point, because such a high percentage of the randomly

chosen factors within the top 50 were important for transformation (and also a few factors with low TFScores as negative controls). The suggestion to test additional factors within the top 20 is unnecessary, as it is virtually certain from the results in the top 50 that almost all will be functionally important. Most importantly, this comment is irrelevant to the central conclusion that a very high percentage of TFs predicted by TFScore to be important actually are."

Random selection is a rather unusual method, which I am not aware of. Sub-sampling out of candidates of a large ranking is fine, but this should be made on a systematic manner (#1,#4,#7,...). The criteria needs to be defined a-priori and described in the text. A pessimistic reader can interpret that all other 30 TFs will fail, which would make the accuracy of the method to be less than 40%.

Authors need to make clear that they use a random selection of top 50 TFs in their manuscript and should include citation of the ENCODE papers using the random selection procedure.

I have not changed anything with respect to the latest comments of “Reviewer” 4. I repeat again my strong disagreement with having Reviewer 4 comment on issues that were never raised during the entire process by the other Reviewers. Moreover, the 2 specific comments are ludicrous and deal with trivial matters that have nothing to do with the conclusions of the paper.

1. *The current manuscript includes no single evidence of the poor performance of footprints on libraries with low number of reads or any evidence that AccessTF performs better than any method for prediction of binding sites. Authors need to remove such statements from the paper or provide experimental data supporting these.*

Based on his/her comments, I strongly suspect that “Reviewer 4” doesn’t understand the difference between DNase hypersensitivity and DNase footprinting. These assays measure different things, even though both use DNase I. For someone who understands these assays, it is obvious on first principles why one needs fewer reads for DNase hypersensitivity measurements than DNase footprints. It has nothing to do with algorithms, but rather the assays themselves. I spelled this out in a previous rebuttal letter, and “Reviewer 4” didn’t address this (probably because of a lack of understanding). In response to the earlier requests, our previous rebuttal letter directly demonstrated that existing DNase footprinting algorithms could not identify TF binding sites with the number of reads we used in the paper. So, we do have experimental data. In this regard, it is important to note that the original Reviewer who requested this computational experiment was satisfied with our response. Lastly, “Reviewer 4” completely misrepresents our statements in the paper. We NEVER compared algorithms in the paper and NEVER said anything about which approach is “better”. Instead, we compared the advantages and disadvantages of footprinting and hypersensitivity ASSAYS.

2. Random selection is a rather unusual method, which I am not aware of. Sub-sampling out of candidates of a large ranking is fine, but this should be made on a systematic manner (#1,#4,#7,...). The criteria needs to be defined a-priori and described in the text. A pessimistic reader can interpret that all other 30 TFs will fail, which would make the accuracy of the method to be less than 40%.

Random selection from a list has been done for years. While I agree that a systematic selection method is better, it doesn't matter if the selection is random. This is the case here, and in this regard, I note that for the vast majority of factors in the top 50 list, we had no idea of which factors would be important and which would not. In other words, cherry picking was impossible. The statement that "a pessimistic reader can interpret that all other 30 TFs will fail, which would make the accuracy of the method to be less than 40%" is so absurd that it calls into question the objectivity of "Reviewer 4".